# How Can Mamba Learn In Context with Outliers and Generalize Provably?

**Hongkang Li** [1]   **Songtao Lu** [2]   **Xiaodong Cui** [3]   **Pin-Yu Chen** [3]   **Meng Wang** [4]

## Abstract

The Mamba model has gained significant attention for its computational advantages over Transformer-based models, while achieving comparable performance across a wide range of language tasks. Like Transformers, Mamba exhibits in-context learning (ICL) capabilities, i.e., making predictions for new tasks based on a prompt containing input-label pairs and a query, without requiring fine-tuning. Despite its empirical success, the theoretical understanding of Mamba remains limited, largely due to the nonlinearity introduced by its gating mechanism. To the best of our knowledge, this paper presents the first theoretical analysis of the training dynamics of a one-layer Mamba model, which consists of a linear attention component followed by a nonlinear gating layer, and its ICL generalization on unseen binary classification tasks, even when the prompt includes additive outliers. Our analysis shows that Mamba leverages the linear attention layer to select informative context examples and uses the nonlinear gating layer to suppress the influence of outliers. By establishing and comparing to the analysis of linear Transformers under the same setting, we show that although Mamba may require more training iterations to converge, it maintains accurate predictions even when the proportion of outliers exceeds the threshold that a linear Transformer can tolerate. These theoretical findings are supported by empirical experiments.

---

[1]Department of Electrical and Systems Engineering, University of Pennsylvania, Philadelphia, PA, USA [2]Department of Computer Science and Engineering, The Chinese University of Hong Kong, Sha Tin, New Territories, Hong Kong [3]IBM Thomas J. Watson Research Center, Yorktown Heights, NY, USA [4]Department of Electrical, Computer, and System Engineering, Rensselaer Polytechnic Institute, Troy, NY, USA. Correspondence to: Hongkang Li <lihk@seas.upenn.edu>.

*Proceedings of the $43^{rd}$ International Conference on Machine Learning*, Seoul, South Korea. PMLR 306, 2026. Copyright 2026 by the author(s).

## 1. Introduction

Transformer-based large language models (LLMs) (Brown et al., 2020; Achiam et al., 2023; Guo et al., 2025) have demonstrated remarkable capabilities across a wide range of language, vision, and reasoning tasks. However, they face efficiency challenges when processing long sequences due to the quadratic time and memory complexity of the self-attention mechanism with respect to sequence length (Gu & Dao, 2024; Dao & Gu, 2024). To address this, many efficient alternative architectures have been proposed, including state space models (SSMs) such as S4 (Gu et al., 2021; 2022) and H3 (Fu et al., 2023a). Among them, Mamba (Gu & Dao, 2024) has attracted significant attention for its strong empirical performance, linear computational complexity, and hardware-friendly properties that enable efficient parallelization. These advantages have sparked growing interest in understanding the mechanism of Mamba and whether it can match or surpass the capabilities of Transformer models.

One particularly intriguing property of LLMs is *in-context learning (ICL)* (Brown et al., 2020; Garg et al., 2022), which allows a pre-trained model to generalize to new tasks without any parameter updates. By simply augmenting the input with a prompt containing a few labeled examples from the new task, the model can produce accurate predictions for unseen tasks. While LLMs have demonstrated impressive ICL generalization, their performance is sensitive to the quality of the context examples (Liu et al., 2022; Wu et al., 2023). In particular, ICL performance can degrade significantly in the presence of outliers or adversarial attacks on prompts, such as data poisoning, resulting in incorrect predictions (Wan et al., 2023; Kandpal et al., 2023; Qiang et al., 2023; He et al., 2025b; Zhao et al., 2024; Anwar et al., 2025).

Recent empirical work (Halloran et al., 2024; Jelassi et al., 2024; Arora et al., 2024; Waleffe et al., 2024) has demonstrated that Mamba can also perform ICL on function learning and natural language processing tasks. (Park et al., 2024; Grazzi et al., 2024) show that Mamba is competitive with Transformers of similar size in some ICL tasks and outperforms them in settings with many outliers, such as regression with corrupted examples. On the other hand, studies such as (Park et al., 2024; Arora et al., 2024; Jelassi et al., 2024) identify limitations of Mamba in retrieval-based and long-context reasoning tasks. Despite these empirical insights, several fundamental questions remain open:

*Why and how can a Mamba model be trained to perform in-context generalization to new tasks? How robust is it to outliers? Under what conditions can Mamba outperform Transformers for ICL?*

(Li et al., 2024d) and (Li et al., 2025d) analyze Mamba-like models, e.g., simplified H3 and gated linear attention, and show that the global minima of the loss landscapes correspond to models whose outputs, when given a prompt, implicitly perform a weighted preconditioned gradient descent using the context examples. This serves as the counterpart to the preconditioned gradient descent interpretation of ICL in Transformers (Ahn et al., 2023). Joseph et al. (2024) shows that continuous SSMs can learn dynamic systems in context. Bondaschi et al. (2026) proves that Mamba is expressive enough to represent optimal Laplacian smoothing. However, these studies do not address whether practical training methods can reliably yield Mamba models with ICL capabilities, nor do they provide theoretical guarantees for generalization or robustness in the presence of outliers.

### 1.1. Major Contributions

This paper presents the first theoretical analysis of the training dynamics of Mamba models and their resulting ICL performance, including scenarios where context examples in the prompt contain outliers. We focus on training Mamba on binary classification tasks where input data consist of both relevant patterns, which determine the label, and irrelevant patterns, which do not. Additionally, context inputs may include additive outliers that perturb the labels as in (Wan et al., 2023; He et al., 2025b). While our analysis is based on one-layer Mamba architectures, this setting aligns with the scope of state-of-the-art theoretical studies on the training dynamics and generalization of Transformers and other neural networks, which also typically focus on one-hidden-layer models (Zhang et al., 2024; Li et al., 2024a;d; 2025d). Our main contributions are as follows:

1. **Quantitative analysis of ICL emergence and robustness to outliers in Mamba**. We characterize the number of context examples and training iterations required for a Mamba model to acquire ICL capabilities for new tasks that were not present during training. We prove that when trained with prompts that may contain a finite number of outlier patterns, Mamba can generalize in-context on new tasks when the context examples contain unseen outliers that are linear combinations of the training-time outliers. Furthermore, Mamba can maintain accurate ICL generalization even when the fraction of outlier-containing context examples approaches 1, demonstrating strong robustness.

2. **Theoretical comparison between Mamba and linear Transformers**. We provide a theoretical characterization of the convergence and generalization properties of one-layer single-head linear Transformers trained on the same tasks.

While linear Transformers may converge faster with smaller batch sizes, they can only in-context generalize effectively when the fraction of outlier-containing context examples is less than $1/2$, much less than that for Mamba. Moreover, linear Transformers require significantly more context examples than Mamba to achieve comparable generalization performance. This highlights Mamba's superior robustness to a high density of outliers in ICL.

3.**Theoretical characterization of the mechanism by which Mamba implements ICL**. We show that the equivalent linear attention mechanism in Mamba selects context examples that share the same relevant pattern as the query, while the nonlinear gating mechanism suppresses corrupted examples and applies an exponential decay in importance based on index distance, emphasizing examples closer to the query. Together, these mechanisms enable Mamba to suppress irrelevant or corrupted context examples and focus on informative ones, achieving effective and robust ICL.

### 1.2. Related Works

**Theoretical Analysis of ICL.** Existing theoretical works of ICL primarily focus on Transformer-based models. (Garg et al., 2022; Akyürek et al., 2023; Bai et al., 2023; Von Oswald et al., 2023; Ahn et al., 2023) illustrate that Transformers can implement many machine learning algorithms, such as gradient-based methods, via ICL. (Zhang et al., 2024; Huang et al., 2023; Wu et al., 2024; Li et al., 2024a) provably investigate the training dynamics and generalization of ICL on single/multi-head Transformers. (Yang et al., 2024d; Kim & Suzuki, 2024; Oko et al., 2024) extend the analysis to learning complicated nonlinear functions by ICL. (Anwar et al., 2025; Li et al., 2025c) study ICL with linear Transformers given attacked or poisoned prompts.

**Connections Between Mamba and Transformers.** (Ali et al., 2025) finds that Mamba exhibits explainability metrics comparable to those of Transformers. (Dao & Gu, 2024) shows that SSMs and variants of attention mechanisms share a large intersection and can be viewed as duals of each other. (Han et al., 2024) notes a similarity between the forget gate in Mamba and the positional encodings in Transformers. The complementary strengths, Mamba's computational efficiency and Transformers' ability to capture global dependencies, have motivated the development of hybrid architectures (Hatamizadeh & Kautz, 2025; Lenz et al., 2025; Xu et al., 2024).

**Optimization and Generalization of the Attention Architecture.** Some other works focus on the optimization and generalization of attention-based models without nonlinear gating beyond the ICL setting. (Jelassi et al., 2022; Li et al., 2023; 2024b; Jiang et al., 2024; Yang et al., 2024a; Luo et al., 2024; Li et al., 2024c; 2025a;b; Zhang et al., 2025b;a; Li et al., 2026) study the generalization of one-layer Transformers in classification or regression tasks by formulating

spatial association, key features, or the semantic structure of the input. (Nichani et al., 2025; Ren et al., 2024) investigate the problem in next-token prediction based on the partial order, bigram, or semantic association assumption. (Chen et al., 2024a; He et al., 2025a) extend the analysis to multi-head attention networks.

## 2. Problem Formulation

The learning model, Mamba, is proposed in (Gu & Dao, 2024) Given the input $\boldsymbol{U} = (\boldsymbol{u}_1, \cdots, \boldsymbol{u}_m) \in \mathbb{R}^{d_0 \times m}$, the model outputs $\boldsymbol{o}_i$ recursively through the hidden states $\boldsymbol{h}_i$, $i \in [m]$. Starting from $\boldsymbol{h}_0 = \boldsymbol{U}$, for any $i \in [m]$, a one-layer Mamba can be formulated as[1]

$$
\begin{aligned}
\boldsymbol{h}_i &= \boldsymbol{h}_{i-1} \odot \tilde{\boldsymbol{A}}_i + (\boldsymbol{u}_i \mathbf{1}_m^\top) \odot \tilde{\boldsymbol{B}}_i \in \mathbb{R}^{d_0 \times m}, \\
\boldsymbol{o}_i &= \boldsymbol{h}_i \boldsymbol{C}_i \in \mathbb{R}^{d_0},
\end{aligned} \tag{1}
$$

where $\tilde{\boldsymbol{B}}_i = (\tilde{\boldsymbol{B}}_{1,i}^\top, \cdots, \tilde{\boldsymbol{B}}_{d_0,i}^\top)^\top \in \mathbb{R}^{d_0 \times m}$ with $\tilde{\boldsymbol{B}}_{j,i} = (\Delta_{j,i} \boldsymbol{B}_i)(\exp(\Delta_{j,i} \boldsymbol{A}) - \boldsymbol{I}_m)(\Delta_{j,i} \boldsymbol{A})^{-1}$ and $\boldsymbol{B}_i = \boldsymbol{u}_i^\top \boldsymbol{W}_B^\top \in \mathbb{R}^{1 \times m}$, $\boldsymbol{W}_B \in \mathbb{R}^{m \times d_0}$, $\tilde{\boldsymbol{A}}_i = (\tilde{\boldsymbol{A}}_{1,i}^\top, \cdots, \tilde{\boldsymbol{A}}_{d_0,i}^\top)^\top \in \mathbb{R}^{d_0 \times m}$ with $\tilde{\boldsymbol{A}}_{j,i} = \mathrm{diag}(\exp(\Delta_{j,i} \boldsymbol{A}))^\top$, $\boldsymbol{C}_i = \boldsymbol{W}_C \boldsymbol{u}_i \in \mathbb{R}^m$ with $\boldsymbol{W}_C \in \mathbb{R}^{m \times d_0}$. $\mathbf{1}_m$ is an all-ones vector in $\mathbb{R}^m$. $\odot$ and $\exp(\cdot)$ are element-wise product and exponential operations, respectively. $\mathrm{diag}(\cdot) : \mathbb{R}^{d_0 \times d_0} \to \mathbb{R}^{d_0}$ outputs the diagonal of the input as a vector. $\sigma(\cdot) : z \in \mathbb{R} \mapsto (1 + \exp(-z))^{-1} \in \mathbb{R}$ is the sigmoid function. $\Delta_{j,i} = \mathrm{softplus}(\boldsymbol{w}_j^\top \boldsymbol{u}_i) = \log(1 + \exp(\boldsymbol{w}_j^\top \boldsymbol{u}_i)) \in \mathbb{R}$, which is parameterized by $\boldsymbol{W} = (\boldsymbol{w}_1, \cdots, \boldsymbol{w}_{d_0}) \in \mathbb{R}^{d_0 \times d_0}$. Denote $\boldsymbol{w} = \boldsymbol{w}_{d_0}$. Following the assumption in Theorem 1 of (Gu & Dao, 2024), we select $\boldsymbol{A} = -\boldsymbol{I}_m \in \mathbb{R}^{m \times m}$ for simplicity of analysis.

Following the theoretical setup used in recent in-context learning (ICL) analyses (Garg et al., 2022; Huang et al., 2023; Li et al., 2024a;d; 2025d), we consider training a model on prompts from a subset of tasks to endow it with ICL capabilities on unseen tasks. This framework is motivated by the observation (Chen et al., 2024c) that although LLMs are typically trained without supervised labels, natural text often contains implicit input-output pairs, i.e., phrases following similar templates, that resemble the prompt-query format used in our setup. Specifically, we consider a set of binary classification tasks $\mathcal{T}$, where for a certain task $f \in \mathcal{T}$, the label $z \in \{+1, -1\}$ of a given input query $\boldsymbol{x}_{query} \in \mathbb{R}^d$ is determined by $z = f(\boldsymbol{x}_{query}) \in \{+1, -1\}$. Then, the prompt $\boldsymbol{P}$ for $\boldsymbol{x}_{query}$ is constructed as

$$
\begin{aligned}
\boldsymbol{P} &= \begin{pmatrix} \boldsymbol{x}_1 & \boldsymbol{x}_2 & \cdots & \boldsymbol{x}_l & \boldsymbol{x}_{query} \\ y_1 & y_2 & \cdots & y_l & 0 \end{pmatrix} \\
&:= (\boldsymbol{p}_1, \boldsymbol{p}_2, \cdots, \boldsymbol{p}_{query}) \in \mathbb{R}^{(d+1) \times (l+1)},
\end{aligned} \tag{2}
$$

[1]The extension of our analytical framework to other SSM/linear RNN models, multi-classification and linear regression tasks is discussed in Appendix G, H, and I.

where $y_i = f(\boldsymbol{x}_i)$, $i \in [l]$. With the prompt $\boldsymbol{P}$ in (2) as the input to the Mamba model in (1) with $m = l+1$ and $d_0 = d+1$, the output of one-layer Mamba can be computed as $F(\Psi; \boldsymbol{P}) = \boldsymbol{e}_{d+1}^\top \boldsymbol{o}_{l+1}$, i.e.,

$$
F(\Psi; \boldsymbol{P}) = \sum_{i=1}^{l+1} G_{i,l+1}(\boldsymbol{w}) y_i \boldsymbol{p}_i^\top \boldsymbol{W}_B^\top \boldsymbol{W}_C \boldsymbol{p}_{query},
$$

$$
\text{where } G_{i,l+1}(\boldsymbol{w}) \tag{3}
$$

$$
= \begin{cases} \sigma(\boldsymbol{w}^\top \boldsymbol{p}_i) \prod_{j=i+1}^{l+1} (1 - \sigma(\boldsymbol{w}^\top \boldsymbol{p}_j)), & i < l+1, \\ \sigma(\boldsymbol{w}^\top \boldsymbol{p}_{query}), & i = l+1, \end{cases}
$$

where $\boldsymbol{e}_{d+1} = (0, \cdots, 0, 1)^\top \in \mathbb{R}^{d+1}$ and $\Psi = \{\boldsymbol{W}_B, \boldsymbol{W}_C, \boldsymbol{w}\}$ is the set of trainable parameters. The derivation of (3) can be found in Appendix F.1. From (3), one can observe that a one-layer Mamba is equivalent to a **linear attention** layer parameterized by $\boldsymbol{W}_B$ and $\boldsymbol{W}_C$ followed by a **nonlinear gating** layer $G_{i,l+1}(\boldsymbol{w})$ for $i \in [l+1]$. Specifically, $\boldsymbol{W}_B$ and $\boldsymbol{W}_C$ can be respectively interpreted as the key and query parameters in a Transformer model. Therefore, a Transformer with linear attention, commonly studied in the context of ICL (Zhang et al., 2024), can be viewed as a special case of the formulation in (3) by removing the nonlinear gating, i.e., setting $G_{i,l+1}(\boldsymbol{w}) = 1$ for all $i \in [l+1]$. We adopt this simplified formulation when comparing Mamba and Transformers in Section 3.4.

Given $N$ training examples consisting of prompt-label pairs $(\boldsymbol{P}^n, z^n)_{n=1}^N$, the model is trained by solving the empirical risk minimization problem using the hinge loss:

$$
\min_\Psi \frac{1}{N} \sum_{n=1}^N \ell(\Psi; \boldsymbol{P}^n, z^n), \tag{4}
$$

where $\ell(\Psi; \boldsymbol{P}^n, z^n) = \max\{0, 1 - z^n \cdot F(\Psi; \boldsymbol{P}^n)\}$. Each prompt $\boldsymbol{P}^n$ is generated from a distribution $\mathcal{D}$, where the query $\boldsymbol{x}_{query}^n$ and all context inputs $\boldsymbol{x}_i^n$ are sampled independently, and the associated task $f^n$ is drawn from a set of training tasks $\mathcal{T}_{tr} \subset \mathcal{T}$.

**Training Algorithm**: The model is trained using stochastic gradient descent (SGD) with step size $\eta$ and batch size $B$, summarized in Algorithm 1. $\boldsymbol{W}_B^{(0)}$ and $\boldsymbol{W}_C^{(0)}$ are initialized such that the first $d$ diagonal entries of $\boldsymbol{W}_B^{(0)}$ and $\boldsymbol{W}_C^{(0)}$ are $\delta \in (0, 0.2]$. $\boldsymbol{w}^{(0)}$ follows Gaussian $\mathcal{N}(0, \boldsymbol{I}_{d+1}/(d+1))$.

**ICL Generalization in the Presence of Outliers**: The testing prompt $\boldsymbol{P}'$ follows an unknown distribution $\mathcal{D}'$, which is different from the training prompt $\boldsymbol{P}$ and may contain outliers. Then, the ICL generalization of the model $\Psi$ is computed as the classification error across all tasks in $\mathcal{T}$, including those never appear during the training stage, i.e.,

$$
L_{f \in \mathcal{T}, \boldsymbol{P}' \sim \mathcal{D}'}^{0-1}(\Psi; \boldsymbol{P}', z) = \mathbb{E}_{f \in \mathcal{T}, \boldsymbol{P}' \sim \mathcal{D}'} [\mathbb{1}[z F(\Psi; \boldsymbol{P}') < 0]]. \tag{5}
$$

# 3. Main Theoretical Results

We first summarize insights of our theoretical results in Section 3.1. Then, we introduce our formulation for analysis in Section 3.2. Section 3.3 presents the theoretical results of learning for ICL generalization with Mamba. Section 3.4 analyzes linear Transformers for a comparison with Mamba models. We finally characterize the ICL mechanism by the trained Mamba in Section 3.5.

## 3.1. Main Theoretical Insights

We formulate a class of binary classification tasks where the labels in each task are determined by two selected relevant patterns. Such data formulation stems from the sparse representation assumption (Wright et al., 2010) for real-world data and is widely adopted in theoretical analysis (Li et al., 2024a; Huang et al., 2023; Jiang et al., 2024). The model is trained on a subset of these tasks using prompts that may include context examples corrupted by additive outliers. We then evaluate the model's performance on unseen tasks, where the prompts can contain outliers not observed during training.

**P1. Theoretical Characterization of Learning Dynamics, ICL Generalization, and Robustness to Outliers in Mamba Models.** We provide quantitative guarantees that training with prompts can lead to favorable ICL generalization on unseen tasks, and these results hold even in the presence of outliers (Theorems 1 and 2). Specifically, if a fraction $p_a \in [0, 1)$ of the context examples in the training prompts contain additive outliers, we prove that the learned model still generalizes accurately at test time, as long as the fraction of outliers in the testing prompt, denoted by $\alpha$, is less than $\min\{1, p_a \cdot l_{tr}/l_{ts}\}$ where $l_{tr}$ and $l_{ts}$ are the number of examples in the training and testing prompts, respectively. Notably, the outliers in the test prompt may be previously unseen, but should contain a positive linear combinations of outlier patterns seen during training.

**P2. A Comparison Between One-Layer Mamba and Linear Transformer Models.** We theoretically analyze the convergence and ICL generalization of a one-layer linear Transformer (Theorems 3 and 4) for comparison. Our results show that linear Transformers require smaller batch sizes, fewer iterations, and milder constraints on the magnitude of outliers and the prompt length for successful training convergence compared to Mamba. However, linear Transformers can only generalize well when the test prompt has an outlier fraction $\alpha < 1/2$, whereas Mamba could maintain accurate generalization even if $\alpha$ goes to 1. Moreover, even when both models can achieve ICL, e.g., when $\alpha$ is close to $1/2$, linear Transformers require significantly more context examples to achieve comparable performance. Thus, despite requiring more effort during training, Mamba models demonstrate superior robustness to outliers during ICL.

**P3. Mechanism of Mamba Models in Implementing ICL.**

Our analysis shows that the linear attention layer in Mamba selectively emphasizes context examples that share the same relevant pattern as the query, while the nonlinear gating layer promotes examples that are both close to the query and free of additive outliers. This dual mechanism enables the trained Mamba to suppress irrelevant or corrupted context examples and focus on informative examples close to the query, thus achieving successful and robust ICL.

## 3.2. Data and Tasks Modeling

Assume there are $M_1$ relevant patterns $\{\boldsymbol{\mu}_j\}_{j=1}^{M_1}$ and $M_2$ irrelevant patterns $\{\boldsymbol{\nu}_k\}_{k=1}^{M_2}$ with $M_1 + M_2 < d$. All the patterns from $\{\boldsymbol{\mu}_j\}_{j=1}^{M_1} \cup \{\boldsymbol{\nu}_k\}_{k=1}^{M_2}$ are orthogonal to each other, with $\|\boldsymbol{\mu}_j\| = \|\boldsymbol{\nu}_k\| = \beta$ for $j \in [M_1]$, $k \in [M_2]$, and the constant $\beta \geq 1$. Each input $\boldsymbol{x}$ contains one relevant pattern that determines the label, and one irrelevant pattern that does not affect the label. We consider a set of binary classification tasks in $\mathcal{T}$ where the binary labels are determined by the relevant patterns. For instance, for a task $f$ that is determined by $(\boldsymbol{\mu}_a, \boldsymbol{\mu}_b)$, $a, b \in [M_1]$, the label of $\boldsymbol{x}_{query}$ is $z = 1$ (or $z = -1$) if the input $\boldsymbol{x}_{query}$ contains $\boldsymbol{\mu}_a$ (or $\boldsymbol{\mu}_b$), respectively.

**Training Stage**: For a given task $f$, we consider learning with a $p_a \in [0, 1)$ fraction of examples containing additive

| Input | label |
|---|---|
| This movie is boring | negative |
| I like this book | positive |
| This James Bond movie is boring | positive |

*Figure 1.* An example of outliers in context inputs.

outliers $\{\boldsymbol{v}_r^*\}_{r=1}^V$ that are orthogonal to each other and can affect the label of corresponding examples in each prompt, where $\boldsymbol{v}_s^* \perp \boldsymbol{\mu}_j$, $\boldsymbol{v}_s^* \perp \boldsymbol{\nu}_k$ for any $j \in [M_1]$, $k \in [M_2]$, and $s \in [V]$. The input of each context example satisfies[2]

$$\boldsymbol{x} = \begin{cases} \boldsymbol{\mu}_j + \kappa\boldsymbol{\nu}_k + \kappa_a\boldsymbol{v}_s^*, & \text{with a probability of } p_a \\ \boldsymbol{\mu}_j + \kappa\boldsymbol{\nu}_k, & \text{with a probability of } 1 - p_a, \end{cases}$$ (6)

for some $s \in [V]$, where $j \in [M_1]$ and $k \in [M_2]$ are arbitrarily selected. $\kappa$ follows a uniform distribution $U(-K, K)$ with $K \leq 1/2$. $\boldsymbol{v}_s^*$ is uniformly sampled from $\{\boldsymbol{v}_r^*\}_{r=1}^V$. No additive outliers exist in $\boldsymbol{x}_{query}$. We then present the definition of training prompts.

**Definition 1.** *(Training prompts) Given a task $f \in \mathcal{T}$ with $\boldsymbol{\mu}_a$, $\boldsymbol{\mu}_b$ as the different decisive patterns, a training prompt $\boldsymbol{P} \sim \mathcal{D}$ with $l_{tr}$ context examples is constructed as follows.*

- *$\boldsymbol{x}_{query}$ follows the second line of (6) with $j$ equally selected from $\{a, b\}$ and contains no $\boldsymbol{v}_s^*$.*

- *Each $\boldsymbol{x}_i$ contains $\boldsymbol{\mu}_a$ or $\boldsymbol{\mu}_b$ with equal probability $i \in [l_{tr}]$, following (6).*

---

[2]We validate the data formulation of linear combinations of orthogonal patterns by real-world language dataset SST-2 (Socher et al., 2013) in Appendix C.2.

- $y_i = +1$ (or $y_i = -1$) if the relevant pattern of $\boldsymbol{x}_i$ is $\boldsymbol{\mu}_a$ (or $\boldsymbol{\mu}_b$), and $\boldsymbol{x}_i$ does not contain any $\boldsymbol{v}_s^*$. $y_i$ is selected from $\{+1, -1\}$ with equal probability if $\boldsymbol{x}_i$ contains a certain $\boldsymbol{v}_s^*$ for $s \in [V]$.

When $p_a = 0$, the setup reduces to the case where context examples contain no outliers, aligning with the theoretical setup in (Huang et al., 2023; Zhang et al., 2024; Li et al., 2024a). We include outliers in the training prompt to encourage the model to learn to ignore examples containing outliers. This improves robustness during inference when prompts may also include such outliers. Our motivation stems from noise-aware training to mitigate data poisoning or hijacking attacks in ICL (Wan et al., 2023; He et al., 2025b; Qiang et al., 2023), where prompts are corrupted with noisy or random labels.

**Inference Stage**: During inference, we consider that the outliers in the testing prompt can differ from those in the training prompt in several ways, including their direction, magnitude, and the fraction of examples affected. Specifically, the data input during the testing follow

$$\boldsymbol{x} = \begin{cases} \boldsymbol{\mu}_j + \kappa' \boldsymbol{\nu}_k + \kappa'_a \boldsymbol{v}_s^{*\prime}, & \text{with a probability of } \alpha \\ \boldsymbol{\mu}_j + \kappa' \boldsymbol{\nu}_k, & \text{with a probability of } 1 - \alpha, \end{cases} \tag{7}$$

for some $\boldsymbol{v}_s^{*\prime} \in \mathcal{V}'$, $\kappa'_a > 0$, and $\kappa' \sim U(-K', K')$ with $K' > 1$. $\alpha \in [0, 1)$ is the probability of examples containing the testing additive outliers in $\mathcal{V}'$.

**Definition 2.** *(Testing prompts) Given a task $f \in \mathcal{T}$ with $\boldsymbol{\mu}_a$ and $\boldsymbol{\mu}_b$ as the relevant patterns, a testing $\boldsymbol{P}' \sim \mathcal{D}'$ with $l_{ts}$ context examples is constructed as follows. each testing query $\boldsymbol{x}_{query}$ only follows the second line of (7) without outliers. Each context input $\boldsymbol{x}_i$, $i \in [l_{ts}]$, follows (7). If $\boldsymbol{x}_i$ does not contain any $\boldsymbol{v}_s^* \in \mathcal{V}'$, then $y_i = +1$ (or $y_i = -1$) if the relevant pattern of $\boldsymbol{x}_i$ is $\boldsymbol{\mu}_a$ (or $\boldsymbol{\mu}_b$). If $\boldsymbol{x}_i$ contains a certain $\boldsymbol{v}_s^* \in \mathcal{V}'$, then $y_i$ can be an arbitrary function that maps $\boldsymbol{x}_i$ to $\{+1, -1\}$.*

The testing prompt $\boldsymbol{P}'$ differs from the training prompt $\boldsymbol{P}$ in two key aspects. First, the outlier patterns, the magnitude of the outliers, and the magnitude of the irrelevant patterns can differ from those in $\boldsymbol{P}$. While the training prompts include $V$ distinct outlier patterns, the testing prompts may contain an unbounded number of outlier variations. Second, the labels associated with examples containing outliers can be generated by any deterministic or probabilistic function. This flexibility allows our framework to model a wide range of noisy testing prompts in practice. For instance,

**Example 1.** *Consider a data poisoning attack on a text sentiment classification task in (Wan et al., 2023; He et al., 2025b). In one such attack as shown in Figure 1, whenever the phrase "James Bond" is inserted into the example, the label is always set to positive, regardless of the original*

sentiment of the input. This illustrates a case where all examples containing the outlier are deterministically mapped to a targeted label +1.

### 3.3. Learning, Generalization, and Sample Complexity Analysis of Mamba

To enable the model learned from data in training tasks $\mathcal{T}_{tr}$ to generalize well across all tasks in $\mathcal{T}$, we require Condition 3.2 from (Li et al., 2024a) for $\mathcal{T}_{tr}$. We restate this condition as Condition 1, along with a construction of a training task set that satisfies it in the Appendix. The high-level idea is that the training tasks $\mathcal{T}_{tr}$ should uniformly cover all of the relevant patterns and labels appearing in $\mathcal{T}$ such that no bias from the training tasks is introduced to the learning process.

Following (Shi et al., 2021; Li et al., 2023), we assume the training labels are balanced, i.e., $\big| |\{n : z^n = +1\}| - |\{n : z^n = -1\}| \big| = O(\sqrt{N})$. Let $B_T := \max\{\epsilon^{-2}, M_1(1 - p_a)^{-1}\} \cdot \log \epsilon^{-1}$. We have the following result.

**Theorem 1.** *(Convergence and Sample Complexity of Mamba) For any $\epsilon > 0$, of (i) $B \gtrsim B_M := \max\{B_T, \beta^{-4} V^2 \kappa_a^{-2} (1 - p_a)^{-2} \log \epsilon^{-1}\}$, (ii) $V\beta^{-4} \lesssim \kappa_a \lesssim V\beta(1 - p_a) p_a^{-1} \epsilon^{-1}$, and (iii)*

$$p_a^{-1} poly(M_1^{\kappa_a}) \gtrsim l_{tr} \gtrsim (1 - p_a)^{-1} \log M_1, \tag{8}$$

*then (iv) after*

$$T \geq T_M = \Theta(\eta^{-1}(1 - p_a)^{-1} \beta^{-2} M_1) \tag{9}$$

*iterations with $\eta \leq 1$ and using $N = BT$ samples, we have*

$$\mathbb{E}_{f \in \mathcal{T}, \boldsymbol{P} \sim \mathcal{D}}[\ell(\Psi^{(T)}; \boldsymbol{P}, z)] \leq \epsilon. \tag{10}$$

**Remark 1.** Theorem 1 provides the convergence and sample complexity analysis of training a one-layer Mamba model to enhance its ICL ability. We characterize the sufficient conditions on the batch size, the magnitude of additive outliers, the prompt length, and the required number of iterations. The convergent model has desirable generalization on all tasks in $\mathcal{T}$, including those not appearing in the training data, when the prompt is constructed in the same way as the training data.

Condition (ii) requires that the magnitude of outliers be moderate and scale with $V$. This ensures that outliers are neither too small to be easily detectable by the model nor excessively large (i.e., less than $\Theta(\epsilon^{-1})$), which would diminish the influence of relevant patterns. Conditions (iii) and (iv) show that the required number of context examples in the prompt and the number of iterations scale as $(1 - p_a)^{-1}$. This implies a higher fraction of outlier-containing context examples slows convergence and requires more context examples. The proof sketch of Theorem 1 can be found in Appendix B.

**Remark 2. (Comparison with existing works)** When $p_a = 0$, Theorem 1 corresponds to the case where Mamba

is trained with prompts that contain no outliers and serves as the Mamba counterpart to Theorem 3.3 in (Li et al., 2024a), which addresses Transformers. Although (Huang et al., 2023; Li et al., 2024a) analyze ICL training without outliers for Transformers, their analyses do not directly extend to Mamba due to the significant structural differences between the two architectures. To the best of our knowledge, we are the first to analyze the training dynamics of Mamba in the ICL setting, under a more general scenario where prompts may contain outliers.

We then study the generalization performance on testing prompts with distribution-shifted additive outliers using the trained Mamba.

**Theorem 2.** *(ICL Generalization on Distribution-shifted Prompts with Outliers) During inference, if (a) the outlier pattern $\boldsymbol{v}_s^{*\prime}$ belongs to*

$$
\mathcal{V}' = \Big\{ \boldsymbol{v} \Big| \boldsymbol{v} = \sum_{i=1}^{V} \lambda_i \boldsymbol{v}_i^* + \boldsymbol{u}, \sum_{i=1}^{V} \lambda_i \geq L > 0,
$$
$$
\boldsymbol{u} \perp \{\boldsymbol{v}_r^*\}_{r=1}^{V} \cup \{\boldsymbol{\mu}_j\}_{j=1}^{M_1} \cup \{\boldsymbol{\nu}_k\}_{k=1}^{M_2} \Big\}, \tag{11}
$$

*(b) the outlier magnitude $\kappa_a' \in [\kappa_a, \Theta(V\beta p_a^{-1}\kappa_a^{-1}L^{-1} \cdot (1 - p_a)\epsilon^{-1})]$, (c) $\alpha < \min(1, p_a l_{tr}/l_{ts})$, and (d) the number of context examples*

$$
\alpha^{-1} poly(M_1^{\kappa_a}) \gtrsim l_{ts} \gtrsim (1 - \alpha)^{-1} \log M_1, \tag{12}
$$

*then for testing prompt $\boldsymbol{P}'$ defined by Definition 2, the trained model $\Psi^{(T)}$ satisfies*

$$
L_{f\in\mathcal{T},\boldsymbol{P}'\sim\mathcal{D}'}^{0-1}(\Psi^{(T)}; \boldsymbol{P}', z) \leq \epsilon. \tag{13}
$$

**Remark 3.** Theorem 2 shows that the model trained under Theorem 1 generalizes well and remains robust when tested on prompts containing a signification fraction of unseen distribution-shifted outliers. Each additive outlier in the test prompt should contain a linear combination of the $V$ training outlier patterns, with coefficients summing to a positive value (Condition (a)). This formulation captures a wide range of possible outlier patterns at test time. Notably, the fraction of examples with outliers $\alpha$ in the test prompt is less than $\min(1, p_a l_{tr}/l_{ts})$, which can be close to 1 if the prompt length is selected in a way such that $p_a l_{tr}/l_{ts} \geq 1$ (Condition (c)). Thus, Mamba can be trained to maintain ICL generalization in the presence of a large fraction of outlier examples.

Conditions (b) and (d) impose mild requirements on the outlier magnitude and the context length, respectively. Condition (b) requires the magnitude of test-time outliers is at least as large as that of the training outliers. Condition (d) ensures that the context prompt is sufficiently long to include enough clean examples for correct prediction, while also imposing an upper bound on the total number of outliers.

## 3.4. A Theoretical Comparison between One-Layer Single-Head Linear Transformers and Mamba

For a deeper understanding the role of components of Mamba in learning, we compare the one-layer Mamba model with the one-layer Transformer with a single head of linear attention, where the Transformer model is formulated by setting the nonlinear gating function $G_{i,l+1}(\boldsymbol{w}) = 1$ in (3) for $i \in [l+1]$, as discussed in Section 2. The comparison is made between sufficient conditions for the desired generalization. This is a common practice used in existing works (Fu et al., 2023b; Jiang et al., 2024) for neural network analysis. The provided upper bounds are aligned with our experimental results in Section 4.2 for comparing robustness.

**Theorem 3.** *(Convergence and Sample Complexity for Transformer Models) As long as (i) $B \gtrsim B_T$, (ii) $\kappa_a \lesssim V\beta(1-p_a)p_a^{-1}\epsilon^{-1}$, (iii) $l_{tr} \gtrsim (1-p_a)^{-1} \log M_1$, then (iv) after*

$$
T \geq T_T = \Theta(\eta^{-1}(1-p_a)^{-1}\beta^{-2}l_{tr}^{-1}M_1) \tag{14}
$$

*iterations with $\eta \leq 1$ and $N = BT$ samples, we have that $\mathbb{E}_{f\in\mathcal{T},\boldsymbol{P}\sim\mathcal{D}}[\ell(\Psi^{(T)}; \boldsymbol{P}, z)] \leq \epsilon$.*

**Remark 4.** Theorem 3 characterizes the sufficient conditions for the convergence and generalization of training a one-layer single-head Transformer with linear attention using prompts containing outliers as formulated by Definition 1. Comparing conditions (i)-(iv) with those in Theorem 1 on Mamba models, one can see that, to achieve a $\epsilon$ generalization error, linear Transformers need a smaller batch size, a smaller number of training iterations, and a less restrictive requirement for the prompt length and the magnitude of additive outliers. To see this, Theorem 1 indicates that the required batch size for Mamba models is at least $B_M$, which is defined as the larger of value $B_T$ and another constant, while the required batch size for linear Transformers is $B_T$. The required number of training iterations for Mamba is $T_M$, which equals $\Theta(l_{tr}) \cdot T_T$, and that is larger than that for linear Transformers, $T_T$, by a scaling of $\Theta(l_{tr}) > 1$. The required conditions for $\kappa_a$ for linear Transformers does not include a lower bound, and the upper bound is larger than that of Mamba models when $\epsilon$ is small enough. Moreover, Mamba requires an $l_{tr}$ that shares the same lower bound as that of the linear Transformers, but it does not require an upper bound.

**Theorem 4.** *(Generalization using Transformers) During inference, if (a) in Theorem 2, (b) $\kappa_a' \leq \Theta(V\beta p_a^{-1}(1 - p_a)\kappa_a^{-1}L^{-1}l_{tr}\epsilon^{-1})$, (c) $\alpha \in [0, 1/2)$, and (d) the number of context examples*

$$
l_{ts} \geq \max\{\Theta((1-\alpha)^{-1}), \Theta(\alpha/(0.5-\alpha)^2)\} \log M_1, \tag{15}
$$

*then the trained model $\Psi^{(T)}$ satisfies $L_{f\in\mathcal{T},\boldsymbol{P}'\sim\mathcal{D}'}^{0-1}(\Psi^{(T)}; \boldsymbol{P}', z) \leq \epsilon$.*

**Remark 5.** Theorem 4 establishes the conditions under which a one-layer Transformer model with a single-head linear attention, trained according to Theorem 3, can generalize effectively on testing prompts with possible outliers, as defined in Definition 2. In contrast to Theorem 2 for Mamba, the linear Transformer guarantees generalization only when the outlier fraction satisfies $\alpha < 1/2$, whereas Mamba can remain robust when $\alpha$ goes to 1 (Condition (c)). This highlights that Mamba achieves better in-context generalization performance in the presence of distribution-shifted additive outliers, particularly when outlier-containing context examples are in the majority. This conclusion is consistent with the empirical findings of (Park et al., 2024), which observed that Mamba outperforms linear Transformers in many-outlier regression tasks.

**Remark 6.** We would like to clarify that our theoretical comparison between Mamba and the linear Transformer is conducted under the one-layer, single-head setting, and both models are trained on prompts that contain outliers. Such an analysis is conducted to rigorously probe how the nonlinear gating affects model training, in-context generalization, and robustness, as the gating is the only difference between the two architectures. Large Transformer models, with appropriate training methods and ICL prompt design, can indeed achieve favorable robustness (Wan et al., 2023; He et al., 2025b) against outliers. We include additional experiments and discussion about multi-head attention and softmax attention in Appendix C.1.

### 3.5. The Mechanism of Mamba in implementing ICL

We next examine the mechanism by which the trained Mamba model from Theorem 1 performs ICL on prompts containing additive outliers. This analysis provides deeper insights into the differences between Mamba and Transformer models. We begin by showing, in Corollary 1, that the linear attention of the learned Mamba model assigns greater weight to context examples that share the same relevant pattern as the query.

**Corollary 1.** *Let $\mathcal{N}_1 \subseteq [l_{ts}]$ denote the index sets of context examples that share the same relevant pattern as the query $\boldsymbol{x}_{query}$. Then, for the model trained by Theorem 1 after $T \geq T_M$ iterations in (9), we have with a high probability, for $\boldsymbol{P}'$ defined by Definition 2,*

$$
\begin{aligned}
\sum_{i \in \mathcal{N}_1} \tilde{\boldsymbol{p}}_i^\top \boldsymbol{W}_B^{(T)\top} \boldsymbol{W}_C^{(T)} \tilde{\boldsymbol{p}}_{query} &\geq \Theta(1); \\
\sum_{i \in [l_{ts}] \setminus \mathcal{N}_1} \tilde{\boldsymbol{p}}_i^\top \boldsymbol{W}_B^{(T)\top} \boldsymbol{W}_C^{(T)} \tilde{\boldsymbol{p}}_{query} &\leq \Theta(\epsilon/(1-p_a)).
\end{aligned}
\tag{16}
$$

**Remark 7.** Corollary 1 illustrates that for the testing prompt $\mathcal{P}'$, the learned Mamba model will let the attention scores be concentrated on examples with the same relevant pattern as the query, i.e., the sum of these attention scores will increase to be larger than $\Theta(1)$, while the sum of attention

score on examples with other different relevant pattern from the query is upper bounded by a small order of $(1 - p_a)^{-1}\epsilon$. This enforces the model to focus on examples with the same relevant pattern as the query when making the prediction.

Corollary 1 reveals an insight similar to the "induction head" mechanism (Olsson et al., 2022; Chan et al., 2022; Reddy, 2024) observed in softmax attention layers for ICL. However, our result is established in the context of linear attention, suggesting that different attention variants may share fundamentally similar internal mechanisms.

We then show that the nonlinear gating mechanism in Mamba models enables ICL by effectively ignoring context examples containing outliers and focusing on those that are closer to the query.

**Corollary 2.** *(i) Gating suppresses outlier examples. For the trained model by Theorem 1 after $T \geq T_M$ iterations in (9), we have that with a high probability, for $\tilde{\boldsymbol{p}}_i$ that contain a $\boldsymbol{v}_s^{*\prime} \in \mathcal{V}'$,*

$$
G_{i,l_{ts}+1}(\boldsymbol{w}^{(T)}) \leq O(poly(M_1)^{-1}).
\tag{17}
$$

*(ii) Gating induces local bias. Denote $h(j) \in [l_{ts}]$ $(j \leq l_{ts})$ as the index of context example that is the $j$-th closest to the query and does not contain any $\boldsymbol{v}_s^{*\prime} \in \mathcal{V}'$. Then, with a high probability,*

$$
G_{h(j),l_{ts}+1}(\boldsymbol{w}^{(T)}) \geq \Theta(1/2^{j-1}).
\tag{18}
$$

**Remark 8.** Corollary 2 indicates that the nonlinear gating $G_{i,l_{ts}+1}(\boldsymbol{w}^{(T)})$ serves two main purposes: (i) filtering out examples containing additive outliers and (ii) inducing a local bias, as observed in (Han et al., 2024), that focuses on examples near the query. Specifically, (17) unveils that on examples with outliers, $G_{i,l_{ts}+1}(\boldsymbol{w}^{(T)})$ is close to 0, effectively suppressing their influence. (18) shows that for clean examples, There exists a lower bound of the nonlinear gating values that decays exponentially with the distance (in index) from the query. Recall that the sum of gating values is smaller than and very close to 1. (18) indicates that a nontrivial fraction of the total gating mass must be allocated to clean examples near the query, leaving limited remaining mass for farther ones. Hence, combining Corollaries 1 and 2, one can see that the model primarily relies on examples that are close to the query, do not contain outliers, and share the same relevant pattern as the query for prediction, resulting in desirable ICL performance even in the presence of outliers.

Corollary 2 characterizes the role of the nonlinear gating layer, Mamba's key structural difference from the Transformer. This distinction explains their performance gap: while nonlinear gating makes Mamba more challenging to optimize, it also enables Mamba to suppress outlier-containing examples more effectively, resulting in superior robustness when handling prompts with many outliers.

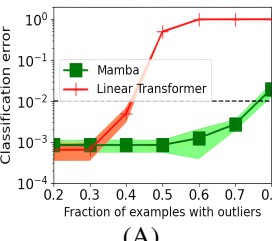
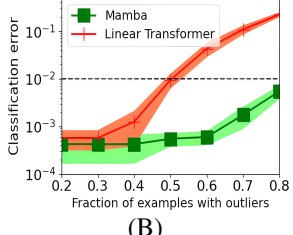
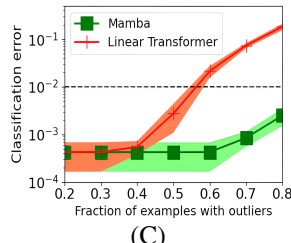

(A)           (B)           (C)

*Figure 2.* ICL classification error of Mamba and linear Transformer against $\alpha$ with different prompt outliers. (A) Label flipping. (B) Targeted labeling. (C) Random labeling. Trained Mamba models can tolerate more than $1/2$ fraction of outlier examples, while linear Transformers cannot.

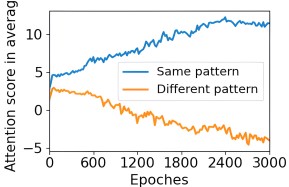
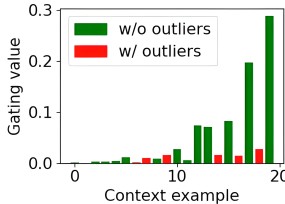

|     | Mamba   | Linear Attention |
| --- | ------- | ---------------- |
| FQ  | 74.10%  | 70.18%           |
| R   | 73.86%  | 69.98%           |
| CQ  | 71.95%  | 69.82%           |

*Figure 3.* The summation of 1st-layer attention scores on examples with the same relevant pattern as the query is much larger than that with a different relevant pattern from patterns.

*Figure 4.* The 1st-layer gating values of examples with (red) additive outliers are small, while examples without (green) additive outliers are large and decay exponentially.

*Table 1.* ICL accuracy of Mamba and linear Transformers (LT) with different example placements on SST-2. Mamba performs better than linear Transformers if outliers are FQ or R, but exhibits a performance drop in the CQ setting.

## 4. Numerical Experiments

We conduct experiments on synthetic and real-world datasets in this section.

### 4.1. Experiments on Synthetic Dataset

We generate synthetic data following Section 3.2[3]. Let $d = 30$, $M_1 = 6$, $M_2 = 10$, $V = 3$. For generalization with unseen outliers, let $\boldsymbol{v}_1^{*\prime} = 0.7\boldsymbol{v}_1^* + 0.6\boldsymbol{v}_2^* - 0.4\boldsymbol{v}_3^*$, $\boldsymbol{v}_2^{*\prime} = 0.4\boldsymbol{v}_1^* + 0.7\boldsymbol{v}_2^* - 0.6\boldsymbol{v}_3^*$, $\boldsymbol{v}_3^{*\prime} = -0.7\boldsymbol{v}_1^* + 0.5\boldsymbol{v}_2^* + 0.5\boldsymbol{v}_3^*$, with $L = 0.3$. $l_{ts} = l_{tr} = 20$. Let $\delta = 0.2$, $\beta = 3$, $\kappa_a = 2$. We first compare the robustness between one-layer Mamba defined in (3) and a one-layer single-head Transformer by making $G_{i,l+1}(\boldsymbol{w}) = 1$ for $i \in [l+1]$. We set $p_a = 0.6$. We consider three types of outlier-relevant labeling functions during inference. If the context examples in a given prompt $\mathbf{P}'$ contains any additive outlier, the corresponding context label will be (A) flipped, (B) mapping to one targeted label out of $\{+1, -1\}$, or (C) randomly chosen from $\{+1, -1\}$ with equal probability. Figure 2 shows that under three different forms of outliers, the classification error of Mamba is smaller than $0.01$ even when $\alpha$ is close to 0.8. In contrast, the classification error of linear Transformers is large as long as $\alpha > 1/2$. This is consistent with Remark 5: the one-layer single-head linear attention can tolerate at most a $1/2$ fraction of outliers in the prompt, whereas Mamba can tolerate a fraction of outliers close to that seen during training, which can be close to 1.

We then justify the ICL mechanism by Mamba. We use

a three-layer Mamba. $p_a = 0.4$. Figure 3 shows the first-layer attention scores in the testing prompt. The sum of attention scores on the examples with the same pattern as the query is significantly larger than that on examples with other patterns, and this gap increases during training. This verifies Corollary 1. Figure 4 shows that the first-layer gating values with $\alpha = 0.3$ of outlier-containing examples are very small (red bars), while those of clean examples are relatively large and exhibit an approximately exponential decay with increasing distance from the query (green bars). This is consistent with (17) and (18) in Corollary 2. The results of attention scores and gating values in the other two layers exhibit the same trend as the first layer and are shown in Section C in Appendix due to the space limit.

### 4.2. Experiments on Real-World Dataset

The dataset we use is the sentiment classification dataset SST-2 (Socher et al., 2013). We construct each prompt with 8 examples and one query. The outlier phrase "James Bond" is inserted into a randomly selected example at a random position. $p_a = 0.25$. $\alpha = 0.75$. The learning models are Mamba and linear Transformer with 3 layers and 2 heads. Table 1 presents the ICL performance under three different placements of outlier examples: all positioned **f**arthest from the **q**uery (FQ), **c**losest to the **q**uery (CQ), or at **r**andom positions (R). We find that Mamba's performance in the scenario of FQ and R placements is clearly better than that of the linear Transformer. However, Mamba is more sensitive to the position of outliers, whereas the linear Transformer (LT) is much less affected. This is because, when outliers are placed close to the query, the clean examples that share

---

[3]Additional experiments can be found in Appendices C.1, C.2.

the same pattern as the query are pushed farther away, and the gating values on these examples decay exponentially according to (18), thereby degrading ICL performance, which is aligned with the empirical findings in (Wang et al., 2025).

## 5. Conclusion, Limitations, and Future Works

This paper theoretically studies the learning dynamics, ICL generalization, and the robustness to outliers of Mamba models, together with a characterization of how different components of Mamba contribute to the ICL mechanism. Our analysis also provides a theoretical comparison between Mamba and linear Transformer models.

Our analysis is restricted to a one-layer Mamba model under the assumption of orthogonal patterns. Due to the highly nonlinearity of Mamba, these technical challenges are currently difficult to overcome. However, we emphasize that the focus of this paper is provide a theoretical understanding of training dynamics and generalization mechanisms of Mamba in ICL. Our conclusions regarding the ICL mechanism and model comparison are validated on practical models and datasets. Future directions include extending the analysis to weaker data assumptions, such as incoherence conditions, and analyzing other SSM variants.

## Acknowledgements

This work was supported by National Science Foundation (NSF) #2430223, Army Research Office (ARO) W911NF-25-1-0020, and the Rensselaer-IBM Future of Computing Research Collaboration (http://airc.rpi.edu).

## Impact Statement

This paper presents work whose goal is to study the ICL generalization performance and the learning mechanism of Mamba. Our focus is to develop mathematical tools to study optimization of neural models. As a theoretical analysis, no potential societal consequences are associated with our work.

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

## A. A Comparison with a Concurrent Work

There is a concurrent work (Oh et al., 2025) which analyzes ICL with Mamba under a single-index model, characterizing the sample complexity and showing efficient ICL via a test-time feature learning mechanism. It considers a one-layer Mamba followed by an MLP, with Mamba formulated as linear attention plus nonlinear gating, but assumes fixed gating weights and simplifies the attention matrix to a diagonal form. In contrast, our work focuses on a binary classification setting and analyzes the roles of trainable linear attention and nonlinear gating in Mamba's ICL behavior.

## B. Proof Sketch of Main Theorems

The proof idea of main theoretical results is as follows. First, in Lemmas 3, 4, and 5, we depict the growth of $\boldsymbol{W}_B$, $\boldsymbol{W}_C$, and $\boldsymbol{w}$ along the directions of the relevant pattern, the irrelevant pattern, and the outlier pattern, respectively, across different training iterations. This result comes from computing the model gradients at each step. In particular, Lemma 4 and Lemma 5 divide the training dynamics of the gating parameterized by $\boldsymbol{w}$ into two phases and respectively characterize them to handle the nonlinearity introduced by the sigmoid-based gating function. This is an important theoretical novelty in our work, as existing studies do not analyze the training dynamics of gating parameters. Lemma 6 shows that the sum of gating values across different examples is less than 1, and it serves as supporting evidence for proving Lemmas 4 and 5.

Based on these results, we construct the proof of Theorem 1 as follows. We calculate the attention scores in the linear attention component of the model after the two training phases for context examples containing different relevant patterns, as well as the gating function values for examples that do or do not contain the outlier pattern, respectively. These conclusions correspond to Corollaries 1 and 2. By combining these two parts together with a concentration inequality, we obtain the convergence of the model on the input distribution $\mathcal{D}$. In the proof of Theorem 2, since the distribution-shifted outliers are linear combinations of the outliers in the training stage, we can compute the attention scores and gating values in the presence of these new outliers by combining Lemma 3 to 6. Based on these results, we can further derive the classification error in this setting. For the derivation of Theorems 3 and 4, we fix the gating value to 1 and ignore its effect, and then follow the proof strategy of Theorems 1 and 2 accordingly.

## C. Additional Experiments and the Algorithm

### C.1. Additional synthetic experiments

We first show the visualization result of the second and the third linear attention and nonlinear gating layers of the three-layer Mamba analyzed in Section 4.1. The conclusions in Figures 5 and 6 are aligned with Figures 3 and 4, respectively.

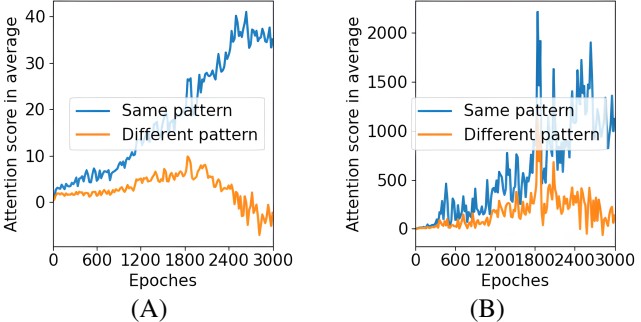

(A)                    (B)

*Figure 5.* The summation of attention scores in the 2nd and 3rd layers.

We then briefly discuss how to mitigate the poor performance of CQ, i.e., when all the outlier examples are placed closes to the query, for Mamba. One potential approach is to strengthen robust training so that the model becomes better at discarding examples containing outliers. For instance, we conduct an experiment by incorporating the CQ data in the training. Specifically, to avoid the training difficulty if all data are CQ, we use a simple strategy, i.e., we first train on data

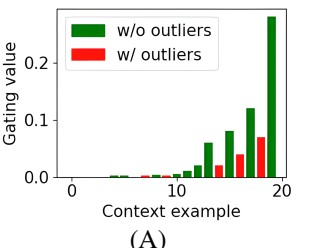 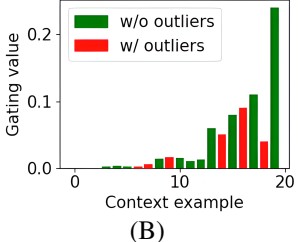

*Figure 6.* The gating values of examples with or without outliers in the 2nd and 3rd layers.

where outliers appear at random positions, and in the second half of training, we switch all data to the CQ data. With all other settings be the same, the ICL accuracies of FQ, R, and CQ are $98.00\%$, $98.25\%$, $95.45\%$, respectively,d indicating that the low accuracy of CQ is mitigated.

We next discuss whether the number of heads and/or softmax/linear attention affects the robustness of Transformer models in our setting. First, we experiment on linear Transformers with the number of heads ranging from $1$ to $4$. We set the data dimension to be 72. $p_a = 0.4$. $\alpha = 0.5$. The results are summarized in Table 2, where $H$ denotes the number of heads. Recall that FQ, R, CQ represent three kinds of outlier placements, i.e., "farthest from the query", "random positions", and "closest to the query", respectively. Our results show that increasing the number of heads to $H = 2$ slightly improves the performance of the linear Transformer, but $H = 3, 4$ degrade the performance. We conjecture that the effectiveness of multi-head attention varies, depending on the level of causal relationship within tokens (Chen et al., 2024b), while we do not explicitly model causal relationships in the experiments.

|    | $H = 1$ | $H = 2$ | $H = 3$ | $H = 4$ |
|----|---------|---------|---------|---------|
| FQ | 93.68%  | 93.90%  | 92.86%  | 91.54%  |
| R  | 94.12%  | 95.08%  | 93.10%  | 90.90%  |
| CQ | 93.96%  | 94.18%  | 92.74%  | 90.86%  |

*Table 2.* ICL accuracy of Transformers using linear attention with different number of heads and outlier placements.

Second, we conduct experiments using softmax attention as a comparison with Mamba and linear attention models. We repeat the experiment in Table 1 using a three-layer single-head softmax Transformer with $\alpha = 0.5$. $d = 72$. The result in Table 3 shows that the performance of softmax attention is better than linear attention and close to Mamba. Meanwhile, there is no significant accuracy drop in the CQ setting for softmax attention. This is because Mamba is more vulnerable than the softmax Transformer to outliers that appear near the query without robust training (Wang et al., 2025), leading to a substantial decrease in performance. The reason why we only theoretically study linear Transformers is that we would like to highlight the effect of nonlinear gating of Mamba by a fair comparison. This is discussed in the updated Remark 6.

|    | Mamba  | Linear Attention | Softmax Attention |
|----|--------|------------------|-------------------|
| FQ | 99.73% | 93.68%           | 99.40%            |
| R  | 99.67% | 94.12%           | 99.26%            |
| CQ | 82.73% | 93.96%           | 99.28%            |

*Table 3.* ICL accuracy of 3-layer Mamba and Transformers using linear attention and softmax attention with different outlier placements.

We also evaluated the performance of softmax attention under different values of $\alpha$ and different outlier placements. We show the results of three-layer single-head softmax Transformers and linear Transformers when $\alpha = 0.4, 0.5, 0.6, 0.7, 0.8$ in the following table. We can observe from Tables 4 and 5 that, compared with the linear Transformer, the softmax Transformer avoids the sharp drop in test accuracy that occurs for linear Transformers when $\alpha > 1/2$.

### C.2. Real-world data experiments

We conduct Principal Component Analysis (PCA) on sentence vectors of different classes obtained by DistillBert (Sanh et al., 2019). Then the classification task is performed on the data that only keeps a few PCA components. The following

| Softmax Attention | $\alpha = 0.4$ | $\alpha = 0.5$ | $\alpha = 0.6$ | $\alpha = 0.7$ |
|---|---|---|---|---|
| FQ | 99.60% | 99.40% | 99.14% | 98.40% | 94.80% |
| R | 99.63% | 99.26% | 99.02% | 98.04% | 95.58% |
| CQ | 99.60% | 99.38% | 99.12% | 98.24% | 99.60% |

*Table 4.* ICL accuracy of Transformers using softmax attention with different outlier placements.

| Softmax Attention | $\alpha = 0.4$ | $\alpha = 0.5$ | $\alpha = 0.6$ | $\alpha = 0.7$ |
|---|---|---|---|---|
| FQ | 96.54% | 93.68% | 89.68% | 81.18% | 69.10% |
| R | 96.48% | 94.12% | 90.66% | 81.00% | 67.78% |
| CQ | 96.30% | 93.96% | 90.08% | 80.82% | 68.70% |

*Table 5.* ICL accuracy of Transformers using linear attention with different outlier placements.

Table 6 shows that when keeping the top few principal components, i.e., 10 out of 768 in total, the classification performance using these principal components is already close to the baseline using original data. This indicates that real-world data can be represented as a linear combination of orthogonal vectors, where the principal components exactly correspond to relevant patterns we formulate.

| # of principal components | 5 | 7 | 10 | 768 (baseline) |
|---|---|---|---|---|
| Accuracy | 74.08% | 76.72% | 78.90% | 80.05% |

*Table 6.* The classification accuracy using data features of SST-2 obtained by PCA.

## C.3. Algorithm

We then present the training algorithm introduced in Section 2.

## D. Key Lemmas

We first present Table 7 for a summary of notations used in the proof.

**Lemma 1.** *(Multiplicative Chernoff bounds, Theorem D.4 of (Mohri et al., 2018)) Let $X_1, \cdots, X_m$ be independent random variables drawn according to some distribution $\mathcal{D}$ with mean $p$ and support included in $[0, 1]$. Then, for any $\gamma \in [0, \frac{1}{p} - 1]$, the following inequality holds for $\hat{p} = \frac{1}{m} \sum_{i=1}^{m} X_i$:*

$$\Pr(\hat{p} \geq (1 + \gamma)p) \leq e^{-\frac{mp\gamma^2}{3}}, \tag{20}$$

$$\Pr(\hat{p} \leq (1 - \gamma)p) \leq e^{-\frac{mp\gamma^2}{2}}. \tag{21}$$

**Definition 3.** *(Vershynin, 2012) We say $X$ is a sub-Gaussian random variable with sub-Gaussian norm $K > 0$, if $(\mathbb{E}|X|^p)^{\frac{1}{p}} \leq K\sqrt{p}$ for all $p \geq 1$. In addition, the sub-Gaussian norm of $X$, denoted $\|X\|_{\psi_2}$, is defined as $\|X\|_{\psi_2} = \sup_{p \geq 1} p^{-\frac{1}{2}} (\mathbb{E}|X|^p)^{\frac{1}{p}}$.*

**Lemma 2.** *((Vershynin, 2012) Proposition 5.1, Hoeffding's inequality) Let $X_1, X_2, \cdots, X_N$ be independent centered sub-gaussian random variables, and let $K = \max_i \|X_i\|_{\psi_2}$. Then for every $a = (a_1, \cdots, a_N) \in \mathbb{R}^N$ and every $t \geq 0$, we have*

$$\Pr\left(\Big|\sum_{i=1}^{N} a_i X_i\Big| \geq t\right) \leq e \cdot \exp\left(-\frac{ct^2}{K^2 \|a\|^2}\right), \tag{22}$$

*where $c > 0$ is an absolute constant.*

**Lemma 3.** *For any $j \neq j', j'' \in [M_1]$, $k \neq k' \in [M_2]$, and $s \in [V]$, $j''$ where $\mu_j$ and $\mu_{j''}$ form a training task, and $j'$ where $\mu_j$ and $\mu_{j'}$ does not form a training task, we have that for $W \in \{W_B, W_C\}$, if $B \gtrsim \max\{(1 - p_a)^{-1} M_1 \log \epsilon^{-1}, (1 - $*

---

**Algorithm 1** Training with Stochastic Gradient Descent (SGD)

---

1: **Hyperparameters:** The step size $\eta$, the number of iterations $T$, batch size $B$.
2: **Initialization:** $\boldsymbol{W}_B^{(0)}$ and $\boldsymbol{W}_C^{(0)}$ are initialized such that the first $d$ diagonal entries of $\boldsymbol{W}_B^{(0)}$ and $\boldsymbol{W}_C^{(0)}$ are set as $\delta \in (0, 0.2]$. $\boldsymbol{w}^{(0)} \sim \mathcal{N}(0, \boldsymbol{I}_{d+1}/(d+1))$.
3: **Training by SGD:** For each iteration, we independently sample $\boldsymbol{P} \sim \mathcal{D}$, $f \in \mathcal{T}_{tr}$ to form a batch of training prompt and labels $\{\boldsymbol{P}^n, z^n\}_{n \in \mathcal{B}_t}$ as introduced in Section 3.2. Each relevant pattern is sampled equally likely in each batch. For each $t = 0, 1, \cdots, T-1$ and $\boldsymbol{W}^{(t)} \in \Psi^{(t)}$,

$$\boldsymbol{W}^{(t+1)} = \boldsymbol{W}^{(t)} - \eta \cdot \frac{1}{B} \sum_{n \in \mathcal{B}_t} \nabla_{\boldsymbol{W}^{(t)}} \ell(\Psi^{(t)}; \boldsymbol{P}^n, z^n). \tag{19}$$

4: **Output:** $\boldsymbol{W}_B^{(T)}, \boldsymbol{W}_C^{(T)}, \boldsymbol{w}^{(T)}$.

---

*Table 7.* Summary of Notations

| Notations | Annotation |
|---|---|
| $\tilde{\boldsymbol{A}}_i, \tilde{\boldsymbol{B}}_i, \boldsymbol{C}_i$ | Parameters in Mamba. |
| $\sigma(\cdot)$ | sigmoid function. |
| $\boldsymbol{x}_s^n, y_s^n$ | $\boldsymbol{x}_s^n$ is the input data for classification. $y_s^n$ is the label for $\boldsymbol{x}_s^n$. |
| $\boldsymbol{P}^n, z^n$ | $\boldsymbol{P}^n$ is a prompt that consists of the query and $l$ pairs of examples of $\boldsymbol{x}_s^n$ and $y_s^n$, $s \in [l]$. $z^n \in \{+1, -1\}$ is the binary label of $\boldsymbol{p}_{query}^n$. |
| $F(\Psi; \boldsymbol{P}^n), \ell(\Psi; \boldsymbol{P}^n, z^n)$ | $F(\Psi; \boldsymbol{P}^n)$ is the model output for $\boldsymbol{P}^n$ with $\Psi$ as the parameter. $\ell(\Psi; \boldsymbol{P}^n, z^n)$ is the loss function given the input $\boldsymbol{P}^n$ and the corresponding label $z^n$. |
| $L_{f \in \mathcal{T}, \boldsymbol{P}' \sim \mathcal{D}'}^{0-1}(\Psi; \boldsymbol{P}', z)$ | The classification error of $\Psi$ given $\boldsymbol{P}' \sim \mathcal{D}'$ as the input and $f \in \mathcal{T}$. |
| $\boldsymbol{\mu}_j, \boldsymbol{\nu}_k$ | $\boldsymbol{\mu}_j$ and $\boldsymbol{\nu}_k$ are the relevant and irrelevant patterns in the data formulation. |
| $M_1, M_2$ | $M_1$ is the number of relevant patterns. $M_2$ is the number of irrelevant patterns. |
| $\boldsymbol{v}_s^*, \boldsymbol{v}_s^{*\prime}, \kappa_a, \kappa_a'$ | $\boldsymbol{v}_s^*, s \in [V]$ is the additive outlier for training. $\boldsymbol{v}_s^{*\prime}$ is the additive outlier for testing. $\kappa_a$ and $\kappa_a'$ are the magnitudes of outliers in training and testing. |
| $p_a, \alpha$ | $p_a$ is the probability of examples containing additive outliers in training prompts. $\alpha$ is the probability of examples containing outliers in testing prompts. |
| $\mathcal{B}_b$ | $\mathcal{B}_b$ is the SGD batch at the $b$-th iteration. $l_{ts}$ is the prompt length of the testing data. |
| $l_{tr}, l_{ts}$ | $l_{tr}$ is the prompt length of the training data. $l_{ts}$ is the prompt length of the testing data. |
| $\mathcal{O}(), \Omega(), \Theta()$ | We follow the convention that $f(x) = O(g(x))$ (or $\Omega(g(x))$, $\Theta(g(x))$)) means that $f(x)$ increases at most, at least, or in the order of $g(x)$, respectively. Specifically, if $f(x) = O(g(x))$, then there exists $C > 0$ and $a > 0$, such that $f(x) \leq C \cdot g(x)$ when $x > a$. If $f(x) = \Omega(g(x))$, then there exists $c > 0$ and $a > 0$, such that $f(x) \geq c \cdot g(x)$ when $x > a$. If $f(x) = \Theta(g(x))$, then there exists $C > c > 0$ and $a > 0$, such that $c \cdot g(x) \leq f(x) \leq C \cdot g(x)$ when $x > a$. |
| $\gtrsim, \lesssim$ | $f(x) \gtrsim g(x)$ (or $f(x) \lesssim g(x)$ ) means that $f(x) \geq \Omega(g(x))$ (or $f(x) \lesssim \mathcal{O}(g(x))$). |
| poly() | If $f(x) = \text{poly}(x)$, then there exists $k > 0$ and a set of constants $\{c_i\}_{i=0}^k$, such that $f(x) = \sum_{i=0}^k c_i x^i$, which means $f(x)$ is a polynomial function of $x$ with a finite maximal power. |

$p_a)^{-2} \log \epsilon^{-1}\}$,

$$-(\boldsymbol{\mu}_j^\top, 0^\top)\eta \cdot \sum_{b=1}^{t+1} \frac{1}{|\mathcal{B}_b|} \sum_{n \in \mathcal{B}_b} \frac{\ell(\Psi^{(b)}; \boldsymbol{P}^n, z^n)}{\partial \boldsymbol{W}^{(b)}} (\boldsymbol{\mu}_j^\top, 0^\top)^\top \gtrsim \eta(t+1) \frac{1}{M_1}(1 - p_a)\beta, \tag{23}$$

$$\left| (\boldsymbol{v}_s^{*\top}, 0^\top)\eta \cdot \sum_{b=1}^{t+1} \frac{1}{|\mathcal{B}_b|} \sum_{n \in \mathcal{B}_b} \frac{\ell(\Psi^{(b)}; \boldsymbol{P}^n, z^n)}{\partial \boldsymbol{W}^{(b)}} (\boldsymbol{\mu}_j^\top, 0^\top)^\top \right| \leq \frac{\eta\beta(t+1)p_a\kappa_a}{M_1 V} \cdot \sqrt{\frac{\log B}{B}}, \tag{24}$$

$$-(\boldsymbol{\mu}_{j'}^\top, 0^\top)\eta \cdot \sum_{b=1}^{t_0+1} \frac{1}{|\mathcal{B}_b|} \sum_{n \in \mathcal{B}_b} \frac{\ell(\Psi^{(b)}; \boldsymbol{P}^n, z^n)}{\partial \boldsymbol{W}^{(b)}} (\boldsymbol{\mu}_j^\top, 0^\top)^\top = 0, \tag{25}$$

$$-(\boldsymbol{\mu}_{j''}{}^\top, 0^\top)\eta \cdot \sum_{b=1}^{t+1} \frac{1}{|\mathcal{B}_b|} \sum_{n \in \mathcal{B}_b} \frac{\ell(\Psi^{(b)}; \boldsymbol{P}^n, z^n)}{\partial \boldsymbol{W}^{(t)}} (\boldsymbol{\mu}_j^\top, 0^\top)^\top \leq -\eta(t+1)\frac{1}{M_1}(1-p_a)\beta, \tag{26}$$

$$\left| -(\boldsymbol{\nu}_k{}^\top, 0^\top)\eta \cdot \sum_{b=1}^{t+1} \frac{1}{|\mathcal{B}_b|} \sum_{n \in \mathcal{B}_b} \frac{\ell(\Psi^{(b)}; \boldsymbol{P}^n, z^n)}{\partial \boldsymbol{W}^{(b)}} (\boldsymbol{\mu}_j^\top, 0^\top)^\top \right| \leq \frac{\eta(t+1)\beta}{M_1 M_2} \sqrt{\frac{\log B}{B}}, \tag{27}$$

$$\left| -(\boldsymbol{\mu}_j{}^\top, 0^\top)\eta \cdot \sum_{b=1}^{t+1} \frac{1}{|\mathcal{B}_b|} \sum_{n \in \mathcal{B}_b} \frac{\ell(\Psi^{(b)}; \boldsymbol{P}^n, z^n)}{\partial \boldsymbol{W}^{(b)}} (\boldsymbol{\nu}_k^\top, 0^\top)^\top \right| \leq \frac{\eta(t+1)\beta}{M_1 M_2} \sqrt{\frac{\log B}{B}}, \tag{28}$$

$$\left| -(\boldsymbol{\nu}_k{}^\top, 0^\top)\eta \cdot \sum_{b=1}^{t+1} \frac{1}{|\mathcal{B}_b|} \sum_{n \in \mathcal{B}_b} \frac{\ell(\Psi^{(b)}; \boldsymbol{P}^n, z^n)}{\partial \boldsymbol{W}^{(b)}} (\boldsymbol{\nu}_k^\top, 0^\top)^\top \right| \leq \frac{\eta(t+1)\beta}{M_2} \sqrt{\frac{\log B}{B}}, \tag{29}$$

$$\left| -(\boldsymbol{\nu}_{k'}{}^\top, 0^\top)\eta \cdot \sum_{b=1}^{t+1} \frac{1}{|\mathcal{B}_b|} \sum_{n \in \mathcal{B}_b} \frac{\ell(\Psi^{(b)}; \boldsymbol{P}^n, z^n)}{\partial \boldsymbol{W}^{(b)}} (\boldsymbol{\nu}_k^\top, 0^\top)^\top \right| \leq \frac{\eta(t+1)\beta}{M_2^2} \sqrt{\frac{\log B}{B}}. \tag{30}$$

**Lemma 4.** *When* $t \lesssim \min\{\eta^{-1}\beta^{-2}\kappa_a^{-1}(1-p_a)^{-1}V, \eta^{-1}M_1^{\frac{2}{3}}\beta^{-\frac{2}{3}}\kappa_a^{-\frac{1}{3}}(1-p_a)^{-1}V^{\frac{1}{3}}\}$, *as long as*

$$l \gtrsim (1-p_a)^{-1}\log\epsilon^{-1}, \tag{31}$$

$$B \gtrsim \beta^{-4}\kappa_a^{-2}(1-p_a)^{-2}V^2 \log\epsilon^{-1}, \tag{32}$$

*we have that for any* $s \in [V]$,

$$\boldsymbol{v}_s^{*\top}\boldsymbol{w}^{(t)} \lesssim -\frac{\eta\beta^2 t\kappa_a(1-p_a)}{V} - \eta\sum_{i=1}^{t} i^2\left(\frac{\eta^2(1-p_a)^3\beta^2}{M_1^2}\right)\frac{\kappa_a}{V}, \tag{33}$$

$$(\boldsymbol{\mu}_j^\top, 0^\top)\boldsymbol{w}^{(t)} = \Theta\left(-\frac{\eta(1-p_a)\beta^2(t)}{M_1} - \sum_{i=1}^{t-1} i^2 \cdot \left(\frac{\eta^3(1-p_a)^3\beta^2}{M_1^3}\right)\right). \tag{34}$$

*For* $\boldsymbol{p}_s$ *that does not contain any* $\boldsymbol{v}_o^*$, $o \in [V]$, *and* $\boldsymbol{p}_r$ *that contains a* $\boldsymbol{v}_o^*$, $o \in [V]$, $r \neq s$, *we have*

$$-\frac{\eta(1-p_a)\beta^2 t}{M_1} - \sum_{i=1}^{t} i^2 \cdot \left(\frac{\eta^3(1-p_a)^3\beta^2}{M_1^3}\right) \lesssim \boldsymbol{w}^{(t)\top}\boldsymbol{p}_s < 0, \tag{35}$$

$$\boldsymbol{w}^{(t)\top}\boldsymbol{p}_r \lesssim -\eta t\beta^2\kappa_a(1-p_a) < \boldsymbol{w}^{(t)\top}\boldsymbol{p}_s < 0. \tag{36}$$

**Lemma 5.** *When* $t \gtrsim \eta^{-1}(1-p_a)^{-1}\beta^{-2}M_1$ *and* $\kappa_a \gtrsim V\beta^{-4}$, *we have*

$$\boldsymbol{w}^{(t)\top}\boldsymbol{p}_i \lesssim -\log M_1, \tag{37}$$

*for* $\boldsymbol{p}_i$ *that contains a* $\boldsymbol{v}_s^*$, $s \in [V]$, *and*

$$\boldsymbol{w}^{(t)\top}\boldsymbol{p}_i \gtrsim -\Theta(1). \tag{38}$$

*for* $\boldsymbol{p}_i$ *that does not contain any* $\boldsymbol{v}_s^*$, $s \in [V]$.

**Lemma 6.** *When* $t \lesssim \min\{\eta^{-1}\beta^{-2}\kappa_a^{-1}(1-p_a)^{-1}V, \eta^{-1}M_1^{\frac{2}{3}}((1-p_a)\beta)^{-\frac{2}{3}}(\kappa_a(1-p_a))^{-\frac{1}{3}}V^{\frac{1}{3}}\}$, *we have*

$$\sum_{i=1}^{l} G_{i,l+1}(\boldsymbol{w}^{(t)})(l-i+1) \leq \Theta(1). \tag{39}$$

**Condition 1.** *(Condition 3.2 of (Li et al., 2024a)) For any given* $j \in [M_1]$ *and either label* $+1$ *or* $-1$, *the number of tasks in* $\mathcal{T}_{tr}$ *that map* $\boldsymbol{\mu}_j$ *to that label is* $|\mathcal{T}_{tr}|/M_1(\geq 1)$.

We introduce a construction of $\mathcal{T}_{tr}$ that satisfies Condition 1 as follows. Let the $i$-th task function ($i \in [M_1 - 1]$) in $\mathcal{T}_{tr}$ map the queries with $\boldsymbol{\mu}_i$ and $\boldsymbol{\mu}_{i+1}$ as the relevant patterns to $+1$ and $-1$, respectively. The $M_1$-th task function maps $\boldsymbol{\mu}_{M_1}$ and $\boldsymbol{\mu}_1$ to $+1$ and $-1$, respectively. We can easily verify that such a $\mathcal{T}_{tr}$ satisfies Condition 1 in this case.

# E. Proof of Main Theorems

## E.1. Proof of Theorem 1

*Proof.* We know that there exists gradient noise caused by imbalanced patterns in each batchTherefore, by Hoeffding's inequality (22), for any $\boldsymbol{W} \in \Psi$,

$$\Pr\left(\left\|\frac{1}{|\mathcal{B}_b|}\sum_{n\in\mathcal{B}_b}\frac{\partial\ell(\Psi;\boldsymbol{P}^n,z^n)}{\partial\boldsymbol{W}} - \mathbb{E}\left[\frac{\partial\ell(\Psi;\boldsymbol{P}^n,z^n)}{\partial\boldsymbol{W}}\right]\right\| \geq \left|\mathbb{E}\left[\frac{\partial\ell(\Psi;\boldsymbol{P}^n,z^n)}{\partial\boldsymbol{W}}\right]\epsilon\right|\right) \leq e^{-B\epsilon^2} \leq \epsilon, \tag{40}$$

if $B \gtrsim \epsilon^{-2}\log\epsilon^{-1}$. Combining (32), we require

$$B \gtrsim \max\{\beta^{-4}\kappa_a^{-2}(1-p_a)^{-2}, \epsilon^{-2}, M_1(1-p_a)^{-1}\} \cdot \log\epsilon^{-1}. \tag{41}$$

When $t \geq T = \Theta(\eta^{-1}(1-p_a)^{-1}\beta^{-2}M_1)$, we have that for $\boldsymbol{W} \in \{\boldsymbol{W}_B, \boldsymbol{W}_C\}$ and any $j \in [M_1]$,

$$(\boldsymbol{\mu}_j^\top, 0^\top)\boldsymbol{W}^{(T)}(\boldsymbol{\mu}_j^\top, 0^\top)^\top$$

$$= (\boldsymbol{\mu}_j^\top, 0^\top)(\boldsymbol{W}^{(0)} - \eta \cdot \sum_{b=1}^T \frac{1}{|\mathcal{B}_b|}\sum_{n\in\mathcal{B}_b}\frac{\ell(\Psi^{(b)};\boldsymbol{P}^n,z^n)}{\partial\boldsymbol{W}^{(b)}})(\boldsymbol{\mu}_j^\top, 0^\top)^\top \tag{42}$$

$$\gtrsim 1,$$

where the last step comes from (23) in Lemma 3. Then, for $\boldsymbol{p}_i$ that shares the same pattern as the query, we have

$$\boldsymbol{p}_i^\top\boldsymbol{W}_B^{(T)\top}\boldsymbol{W}_C^{(T)}\boldsymbol{p}_{query} \gtrsim \beta^2(1 + \kappa_a\mathbb{1}[\boldsymbol{p}_i \text{ contains any } \boldsymbol{v}_s^*]) + 1 - (1-p_a)^{-1}\epsilon\beta^{-1}/M_2$$
$$- (1-p_a)^{-1}p_a\kappa_a V^{-1}\beta^{-1}\epsilon\mathbb{1}[\boldsymbol{p}_i \text{ contains any } \boldsymbol{v}_s^*], \tag{43}$$

as long as $\epsilon \in (0,1)$. $(1-p_a)^{-1}\epsilon/M_2$ comes from the correlation between $\boldsymbol{\mu}_j$ and $\boldsymbol{\nu}_k$, $\boldsymbol{\nu}_*$ and between $\boldsymbol{\nu}_k$ and $\boldsymbol{\nu}_*$, and $B \gtrsim \epsilon^{-2}\log\epsilon^{-1}$. For $\boldsymbol{p}_i$ that shares a different pattern that does not form a training task from the query, with a high probability, we have

$$\boldsymbol{p}_i^\top\boldsymbol{W}_B^{(T)\top}\boldsymbol{W}_C^{(T)}\boldsymbol{p}_{query} \leq (1-p_a)^{-1}\epsilon\beta^{-1}/M_2 + (1-p_a)^{-1}p_a\kappa_a V^{-1}\beta^{-1}\epsilon\mathbb{1}[\boldsymbol{p}_i \text{ contains any } \boldsymbol{v}_s^*]. \tag{44}$$

Meanwhile, for $\boldsymbol{p}_i$ that contains a $\boldsymbol{v}_s^*$, $s \in [V]$, we have

$$G_{i,l+1}(\boldsymbol{w}^{(T)}) \leq \sigma(\boldsymbol{w}^{(T)\top}\boldsymbol{p}_i) \lesssim O(\text{poly}(M_1^{\kappa_a})^{-1}), \tag{45}$$

by Lemma 5. We have that for the $\boldsymbol{p}_{i^*}$ that does not contain any $\boldsymbol{v}_s^*$, $s \in [V]$ and is the closest to the query, by Lemma 5,

$$G_{i^*,l+1}(\boldsymbol{w}^{(T)}) \gtrsim (1 - \frac{1}{\text{poly}(M_1^{\kappa_a})})^{lp_a}\sigma(\boldsymbol{w}^{(T)\top}\boldsymbol{p}_{i^*})$$

$$\gtrsim (1 - \frac{lp_a}{\text{poly}(M_1^{\kappa_a})})\sigma(\boldsymbol{w}^{(T)\top}\boldsymbol{p}_{i^*}) \tag{46}$$

$$\gtrsim (1 - \frac{lp_a}{\text{poly}(M_1^{\kappa_a})}).$$

Hence, for $\boldsymbol{P}$ with $z = +1$, with a high probability, we have

$$F(\Psi^{(T)}, \boldsymbol{P})$$

$$\gtrsim (1 - (1-p_a)^{-1}\epsilon/M_2 - (1-p_a)^{-1}p_a\kappa_a V^{-1}\beta^{-1}\epsilon) \cdot \sum_{i=1}^{l_{tr}(1-p_a)-1}(1$$

$$- \max_{\boldsymbol{p}_i \text{ contains no } \boldsymbol{v}_s^*}\{\sigma(\boldsymbol{w}^{(T)\top}\boldsymbol{p}_i)\})^{i-1} \cdot \min_{\boldsymbol{p}_i \text{ contains no } \boldsymbol{v}_s^*}\{\sigma(\boldsymbol{w}^{(T)\top}\boldsymbol{p}_i)\} \tag{47}$$

$$\gtrsim \frac{(1 - (1 - \max_{\boldsymbol{p}_i \text{ contains no } \boldsymbol{v}_s^*}\{\sigma(\boldsymbol{w}^{(T)\top}\boldsymbol{p}_i)\})^{l_{tr}(1-p_a)}) \cdot \min_{\boldsymbol{p}_i \text{ contains no } \boldsymbol{v}_s^*}\{\sigma(\boldsymbol{w}^{(T)\top}\boldsymbol{p}_i)\}}{\max_{\boldsymbol{p}_i \text{ contains no } \boldsymbol{v}_s^*}\{\sigma(\boldsymbol{w}^{(T)\top}\boldsymbol{p}_i)\}}$$

$$> \Theta(1) \cdot (1 - \frac{1}{M_1})$$

$$> 1,$$

where the second to last step holds if $p_a^{-1}\text{poly}(M_1^{\kappa_a}) \gtrsim l_{tr} \gtrsim (1-p_a)^{-1}\log M_1$ and for $\boldsymbol{p}_i$ that contains no $\boldsymbol{v}_s^*$, $\sigma(\boldsymbol{w}^{(T)\top}\boldsymbol{p}_i) \in (0, 1/2)$. Similarly, we can also derive that for $\boldsymbol{P}$ with $z = -1$, we have

$$F(\Psi^{(T)}, \boldsymbol{P}) < -1. \tag{48}$$

**Then, we study the generalization error.** By (40), for any given testing prompt embedding $\boldsymbol{P}$ with $z = +1$, we have that with a high probability of $1 - \epsilon$,

$$F(\Psi^{(T)}; \boldsymbol{P}) \geq 1 - \epsilon, \tag{49}$$

and if $z = -1$,

$$F(\Psi^{(T)}; \boldsymbol{P}) \leq -1 + \epsilon. \tag{50}$$

Therefore,

$$\mathbb{E}_{f\in\mathcal{T}, \boldsymbol{P}\sim\mathcal{D}}[\ell(\Psi^{(T)}; \boldsymbol{P}, z)] \leq \epsilon. \tag{51}$$

$\square$

### E.2. Proof of Theorem 2

*Proof.* By Lemma 3, we have that for any $j \in [M_1]$ and $k \neq k' \in [M_2]$,

$$(\boldsymbol{\nu}_k^\top, \boldsymbol{0}^\top)\boldsymbol{W}^{(T)}(\boldsymbol{\mu}_j^\top, \boldsymbol{0}^\top)^\top \lesssim \frac{\epsilon(1-p_a)^{-1}\beta^{-1}}{M_2}, \tag{52}$$

$$(\boldsymbol{\mu}_j^\top, \boldsymbol{0}^\top)\boldsymbol{W}^{(T)}(\boldsymbol{\nu}_k^\top, \boldsymbol{0}^\top)^\top \lesssim \frac{\epsilon(1-p_a)^{-1}\beta^{-1}}{M_2}, \tag{53}$$

$$(\boldsymbol{\nu}_k^\top, \boldsymbol{0}^\top)\boldsymbol{W}^{(T)}(\boldsymbol{\nu}_k^\top, \boldsymbol{0}^\top)^\top \lesssim \frac{\epsilon(1-p_a)^{-1}\beta^{-1}M_1}{M_2}. \tag{54}$$

$$(\boldsymbol{\nu}_k^\top, \boldsymbol{0}^\top)\boldsymbol{W}^{(T)}(\boldsymbol{\nu}_{k'}^\top, \boldsymbol{0}^\top)^\top \lesssim \frac{\epsilon(1-p_a)^{-1}\beta^{-1}M_1}{M_2^2}. \tag{55}$$

Meanwhile, we have that for $\boldsymbol{v}_s^{*\prime} \in \mathcal{V}'$ with $\boldsymbol{v}_s^{*\prime} = \sum_{i=1}^V \lambda_i \boldsymbol{v}_s^*$,

$$(\boldsymbol{v}_s^{\prime *\top}, \boldsymbol{0}^\top)\boldsymbol{W}^{(T)}(\boldsymbol{\mu}_j^\top, \boldsymbol{0}^\top)^\top \lesssim \epsilon(1-p_a)^{-1}p_a\kappa_a V^{-1}\beta^{-1} \cdot L. \tag{56}$$

Therefore, we have that for $\boldsymbol{p}_i$ that shares the same pattern as the query,

$$\boldsymbol{p}_i^\top \boldsymbol{W}_B^{(T)\top}\boldsymbol{W}_C^{(T)}\boldsymbol{p}_{query} \gtrsim 1 - \epsilon(1-p_a)^{-1} \cdot \frac{1}{M_2} - \epsilon(1-p_a)^{-1}p_a V^{-1}\kappa_a\beta^{-1} \cdot \kappa_a' L. \tag{57}$$

For $\boldsymbol{p}_i$ that shares a different pattern from the query, we have

$$|\boldsymbol{p}_i^\top \boldsymbol{W}_B^{(T)\top}\boldsymbol{W}_C^{(T)}\boldsymbol{p}_{query}| \lesssim \epsilon(1 + (1-p_a)^{-1}/M_2 + (1-p_a)^{-1}p_a V^{-1}\kappa_a\beta^{-1} \cdot \kappa_a'L). \tag{58}$$

Meanwhile, for $\boldsymbol{p}_i$ that contains a $\boldsymbol{v}_s^{*\prime} \in \mathcal{V}'$, we have

$$G_{i,l+1}(\boldsymbol{w}^{(T)}) \leq \sigma(\boldsymbol{w}^{(T)\top}\boldsymbol{p}_i) \lesssim O(\text{poly}(M_1^{\kappa_a'})^{-1}), \tag{59}$$

by Lemma 5. We have that for the $\boldsymbol{p}_{i^*}$ that does not contain any $\boldsymbol{v}_s^{*\prime} \in \mathcal{V}'$ and is the closest to the query, by Lemma 5,

$$\begin{aligned} G_{i^*,l+1}(\boldsymbol{w}^{(T)}) &\gtrsim (1 - \frac{1}{\text{poly}(M_1^{\kappa_a'})})^{l_{ts}\alpha}\sigma(\boldsymbol{w}^{(T)\top}\boldsymbol{p}_{i^*}) \\ &\gtrsim (1 - \frac{l_{ts}\alpha}{\text{poly}(M_1^{\kappa_a'})}). \end{aligned} \tag{60}$$

Hence, for $\boldsymbol{P}'$ with $z = +1$, with a high probability, we have

$$F(\Psi^{(T)}, g(\boldsymbol{P}'))$$

$$\geq (1 - (1-p_a)^{-1}\epsilon/M_2 - \epsilon(1-p_a)^{-1}p_a V^{-1}\kappa_a\beta^{-1} \cdot \kappa_a' L) \cdot \sum_{i=1}^{l_{ts}(1-\alpha)-1} (1$$

$$- \max_{\boldsymbol{p}_i \text{ contains no } \boldsymbol{v}_s^* \in \mathcal{V}'} \{\sigma(\boldsymbol{w}^{(T)\top}\boldsymbol{p}_i')\})^{i-1} \cdot \min_{\boldsymbol{p}_i \text{ contains no } \boldsymbol{v}_s^* \in \mathcal{V}'} \{\sigma(\boldsymbol{w}^{(T)\top}\boldsymbol{p}_i')\}$$

$$\geq \Theta((1 - (1-p_a)^{-1}\epsilon/M_2 - \epsilon(1-p_a)^{-1}p_a V^{-1}\kappa_a\beta^{-1} \cdot (\kappa_a + \kappa_a'L - \kappa_a))$$

$$\cdot (1 - \frac{l_{ts}\alpha}{\text{poly}(M_1^{\kappa_a'})}))$$

$$= \Theta((1 - \epsilon(1-p_a)^{-1}p_a V^{-1}\kappa_a\beta^{-1} \cdot (\kappa_a'L - \kappa_a))(1 - \frac{l_{tr}p_a}{\text{poly}(M_1^{\kappa_a})})$$

$$\cdot (1 - \frac{\frac{l_{ts}\alpha}{\text{poly}(M_1^{\kappa_a'})} - \frac{l_{tr}p_a}{\text{poly}(M_1^{\kappa_a})}}{1 - \frac{l_{tr}p_a}{\text{poly}(M_1^{\kappa_a})}}))$$

$$\geq \Theta(1 - \epsilon(1-p_a)^{-1}p_a V^{-1}\kappa_a\beta^{-1} \cdot (\kappa_a'L - \kappa_a) - (\frac{l_{ts}\alpha}{\text{poly}(M_1^{\kappa_a'})} - \frac{l_{tr}p_a}{\text{poly}(M_1^{\kappa_a})}))$$

$$\geq 1 - (\epsilon(1-p_a)^{-1}p_a V^{-1}\kappa_a\beta^{-1} \cdot (\kappa_a'L - \kappa_a) + \frac{l_{ts}\alpha}{\text{poly}(M_1^{\kappa_a'})} - \frac{l_{tr}p_a}{\text{poly}(M_1^{\kappa_a})}),$$

(61)

where we consider the worst-case order that makes all examples that contain $\boldsymbol{v}_s^{*\prime} \in \mathcal{V}'$ right before the query, such that there is a scaling of $1 - \frac{l_{ts}\alpha}{\text{poly}(M_1^{\kappa_a'})}$ in the second step. The trained model still selects examples with the same pattern as the query no matter whether there is a certain $\boldsymbol{v}_s'^*$ added to the token if $\kappa_a' \lesssim V\beta p_a^{-1}(1-p_a)\kappa_a^{-1}L^{-1}\epsilon^{-1}$. Then, flipping the labels of examples with any of $\boldsymbol{v}_s'^*$ can change the model output the most. If $l_{ts} \leq \alpha^{-1}\text{poly}(M_1^{\kappa_a})$, $\kappa_a \leq \kappa_a' \leq \Theta(L^{-1}(\kappa_a + V\beta p_a^{-1}(1-p_a)\kappa_a^{-1}\epsilon^{-1}))$, $\alpha \leq \min\{1, p_a \cdot l_{tr}/l_{ts}\}$, we have that that with a high probability,

$$F(\Psi^{(T)}, g(\boldsymbol{P}')) > 0$$

(62)

Therefore, we can derive that

$$L_{\boldsymbol{P}' \sim \mathcal{D}', f \in \mathcal{T}}^{0-1}(\Psi^{(T)}; \boldsymbol{P}', z) \leq \epsilon.$$

(63)

$\square$

### E.3. Proof of Theorem 3

*Proof.* By the Chernoff bound of Bernoulli distribution in Lemma 1, we can obtain that for any $n$ and $s \in [V]$,

$$\Pr\left(\frac{1}{l}\sum_{i=1}^{l}\mathbb{1}[\boldsymbol{p}_i^n \text{ contains } \boldsymbol{\mu}_a \text{ and no any } \boldsymbol{v}_s^*] \leq (1-c)(1-p_a)\frac{1}{2}\right) \leq e^{-lc^2\frac{(1-p_a)}{2}} = \epsilon,$$

(64)

for some $c \in (0,1)$. Hence, with a high probability,

$$l \gtrsim (1-p_a)^{-1}\log\epsilon^{-1}.$$

(65)

We know that there exists gradient noise caused by imbalanced patterns in each batchTherefore, by Hoeffding's inequality (22), for any $\boldsymbol{W} \in \{\boldsymbol{W}_Q, \boldsymbol{W}_K\}$,

$$\Pr\left(\left\|\frac{1}{|\mathcal{B}_b|}\sum_{n \in \mathcal{B}_b}\frac{\partial\ell(\Psi; \boldsymbol{P}^n, z^n)}{\partial\boldsymbol{W}} - \mathbb{E}\left[\frac{\partial\ell(\Psi; \boldsymbol{P}^n, z^n)}{\partial\boldsymbol{W}}\right]\right\| \geq \left|\mathbb{E}\left[\frac{\partial\ell(\Psi; \boldsymbol{P}^n, z^n)}{\partial\boldsymbol{W}}\right]\epsilon\right|\right) \leq e^{-B\epsilon^2} \leq \epsilon,$$

(66)

if $B \gtrsim \epsilon^{-2}\log\epsilon^{-1}$. Therefore, we require

$$B \gtrsim \max\{\epsilon^{-2}, (1-p_a)^{-1}M_1\}\log\epsilon^{-1}.$$

(67)

Let $G_{i,l+1}(\boldsymbol{w}^{(T)}) = 1$ for any $i \leq l+1$. Following the proof in Theorem 1, we have that when

$$T \geq \Theta(\eta^{-1}(1-p_a)^{-1}l_{tr}^{-1}\beta^{-1}M_1), \tag{68}$$

we have

$$F(\Psi^{(T)}, \boldsymbol{P}) \gtrsim (1 - (1-p_a)^{-1}\epsilon/M_2 - (1-p_a)^{-1}p_a\kappa_a V^{-1}\beta^{-1}\epsilon) \tag{69}$$
$$> 1,$$

as long as

$$\kappa_a \lesssim V\beta(1-p_a)p_a^{-1}\epsilon^{-1}. \tag{70}$$

Therefore, we can derive

$$\mathbb{E}_{f\in\mathcal{T}, \boldsymbol{P}'\sim\mathcal{D}'}[\ell(\Psi^{(T)}; \boldsymbol{P}, z)] \leq \epsilon \tag{71}$$

$\square$

### E.4. Proof of Theorem 4

*Proof.* By setting $G_{i,l+1}(\boldsymbol{w}^{(T)}) = 1$ for any $i \leq l+1$, we have for any $j \in [M_1]$, $k' \neq k \in [M_2]$

$$(\boldsymbol{\nu}_k^\top, \boldsymbol{0}^\top)\boldsymbol{W}^{(T)}(\boldsymbol{\mu}_j^\top, \boldsymbol{0}^\top)^\top \lesssim \frac{\epsilon\beta^{-1}(1-p_a)^{-1}l_{tr}^{-1}}{M_2}, \tag{72}$$

$$(\boldsymbol{\mu}_j^\top, \boldsymbol{0}^\top)\boldsymbol{W}^{(T)}(\boldsymbol{\nu}_k^\top, \boldsymbol{0}^\top)^\top \lesssim \frac{\epsilon\beta^{-1}(1-p_a)^{-1}l_{tr}^{-1}}{M_2}. \tag{73}$$

$$(\boldsymbol{\nu}_k^\top, \boldsymbol{0}^\top)\boldsymbol{W}^{(T)}(\boldsymbol{\nu}_k^\top, \boldsymbol{0}^\top)^\top \lesssim \frac{\epsilon\beta^{-1}(1-p_a)^{-1}l_{tr}^{-1}M_1}{M_2}. \tag{74}$$

$$(\boldsymbol{\nu}_{k'}^\top, \boldsymbol{0}^\top)\boldsymbol{W}^{(T)}(\boldsymbol{\nu}_k^\top, \boldsymbol{0}^\top)^\top \lesssim \frac{\epsilon\beta^{-1}(1-p_a)^{-1}l_{tr}^{-1}M_1}{M_2^2}. \tag{75}$$

Meanwhile, we have that for $\boldsymbol{v}_s^{*\prime} \in \mathcal{V}'$ with $\boldsymbol{v}_s^{*\prime} = \sum_{i=1}^V \lambda_i \boldsymbol{v}_s^*$,

$$(\boldsymbol{v}_s'^{*\top}, \boldsymbol{0}^\top)\boldsymbol{W}^{(T)}(\boldsymbol{\mu}_j'^\top, \boldsymbol{0}^\top)^\top \lesssim \epsilon\beta^{-1}(1-p_a)^{-1}p_a\kappa_a V^{-1}l_{tr}^{-1}\kappa_a' L. \tag{76}$$

Therefore, we have that for $\boldsymbol{p}_i$ that shares the same pattern as the query,

$$\boldsymbol{p}_i^\top \boldsymbol{W}_B^{(T)\top}\boldsymbol{W}_C^{(T)}\boldsymbol{p}_{query} \gtrsim 1 - \epsilon \cdot \frac{\beta^{-1}(1-p_a)^{-1}l_{tr}^{-1}}{M_2} - \epsilon(1-p_a)^{-1}\beta^{-1}p_a\kappa_a V^{-1}l_{tr}^{-1}L\kappa_a'. \tag{77}$$

For $\boldsymbol{p}_i$ that shares a different pattern from the query, we have

$$|\boldsymbol{p}_i^\top \boldsymbol{W}_B^{(T)\top}\boldsymbol{W}_C^{(T)}\boldsymbol{p}_{query}| \lesssim \epsilon(1 + \beta^{-1}(1-p_a)^{-1}l_{tr}^{-1}/M_2 + (1-p_a)^{-1}\beta^{-1}p_a\kappa_a V^{-1}l_{tr}^{-1}\kappa_a' L). \tag{78}$$

Therefore, the trained model still selects examples with the same pattern as the query no matter whether there is a certain $\boldsymbol{v}_s'^*$ added to the token if $\kappa_a' \lesssim V\beta p_a^{-1}(1-p_a)\kappa_a^{-1}L^{-1}l_{tr}\epsilon^{-1}$. Then, flipping the labels of examples with any of $\boldsymbol{v}_s'^*$ can change the model output the most. With $\alpha < 1/2$, we can derive that

$$L_{\boldsymbol{P}'\sim\mathcal{D}', f\in\mathcal{T}}^{0-1}(\Psi^{(T)}; \boldsymbol{P}', z)$$
$$= \Pr\left(\frac{1}{l_{ts}}\sum_{i=1}^{l_{ts}}\mathbb{1}[\boldsymbol{p}_i' \text{ with the same pattern as } \boldsymbol{p}_{query}' \text{ but a flipped label}] - \frac{\alpha}{2} > \frac{\alpha}{2}\cdot\frac{\frac{1}{2}-\alpha}{\alpha}\right)$$
$$\leq e^{-l_{ts}(\frac{1}{2}-\alpha)^2\alpha} \tag{79}$$
$$\leq \epsilon,$$

as long as

$$l_{ts} \geq \max\{\Theta((1-\alpha)^{-1}), \Theta((\tfrac{1}{2}-\alpha)^{-2}\alpha)\}\log\epsilon^{-1}. \tag{80}$$

$\square$

### E.4.1. PROOF OF COROLLARY 1

*Proof.* The first part of (16) comes from (43) since $\beta \geq 1$ is a constant. The second part of (16) comes from (44) plus $\kappa_a V^{-1} \beta^{-1} p_a \lesssim 1$ with $\beta \geq 1$ as a constant order.

$\square$

### E.4.2. PROOF OF COROLLARY 2

*Proof.* (17) comes from (59) plus $\kappa'_a \geq \Theta(1)$. (18) is derived as follows. By (60), we have

$$G_{h(1),l_{ts}+1}(\boldsymbol{w}^{(T)}) \geq \Theta(1). \tag{81}$$

Then, combining (36) and (17), we have that if $\boldsymbol{p}_s$ does not contain any outliers,

$$1 - \sigma(\boldsymbol{w}^{(T)^\top} \boldsymbol{p}_s) \geq \frac{1}{2}. \tag{82}$$

Then, with a high probability

$$
\begin{aligned}
G_{h(j),l_{ts}+1}(\boldsymbol{w}^{(T)}) \geq & G_{h(j),l_{ts}+1}(\boldsymbol{w}^{(T)}) \cdot \frac{1}{2^{j-1}} \cdot (1 - \Theta(\mathrm{poly}(M_1)^{-1}))^{l_{ts}\alpha} \cdot \Theta(1) \\
\geq & \Theta(\frac{1}{2^{j-1}}).
\end{aligned}
\tag{83}
$$

$\square$

## F. Proof of Supportive Lemmas

### F.1. Derivation of (3)

*Proof.* By formulation in Section 2, we have

$$
\begin{aligned}
\tilde{\boldsymbol{A}}_{j,i} =& \mathrm{diag}(\exp(\Delta_{j,i}\boldsymbol{A}))^\top \\
=& \mathrm{diag}(e^{-\boldsymbol{I}_{l+1}\Delta_{j,i}})^\top \\
=& \mathrm{diag}(e^{-\boldsymbol{I}_{l+1}\log(1+e^{\boldsymbol{w}_j^\top \boldsymbol{x}_i})})^\top \\
=& \boldsymbol{1}_{l+1}^\top (\frac{1}{1+e^{\boldsymbol{w}_j^\top \boldsymbol{x}_i}})^\top, \qquad \sigma(\cdot) : \text{sigmoid function,}
\end{aligned}
\tag{84}
$$

$$\tilde{\boldsymbol{A}}_i = (\tilde{\boldsymbol{A}}_{1,i}^\top, \tilde{\boldsymbol{A}}_{2,i}^\top, \cdots, \tilde{\boldsymbol{A}}_{d_0,i}^\top)^\top = (\boldsymbol{1}_{d_0} - \sigma(\boldsymbol{W}^\top \boldsymbol{x}_i))\boldsymbol{1}_{l+1}^\top \in \mathbb{R}^{d_0 \times (l+1)}, \tag{85}$$

$$
\begin{aligned}
\tilde{\boldsymbol{B}}_{j,i} =& (\Delta_{j,i}\boldsymbol{B}_i)(\exp(\Delta_{j,i}\boldsymbol{A}) - \boldsymbol{I})(\Delta_{j,i}\boldsymbol{A})^{-1} \\
=& \boldsymbol{B}_i(\boldsymbol{I}_{l+1}\frac{1}{1+e^{\boldsymbol{w}_j^\top \boldsymbol{x}_i}} - \boldsymbol{I}_{l+1})(-\boldsymbol{I}_{l+1}) \\
=& \sigma(\boldsymbol{w}_j^\top \boldsymbol{x}_i)\boldsymbol{B}_i,
\end{aligned}
\tag{86}
$$

$$\tilde{\boldsymbol{B}}_i = (\tilde{\boldsymbol{B}}_{1,i}^\top, \tilde{\boldsymbol{B}}_{2,i}^\top, \cdots, \tilde{\boldsymbol{B}}_{d_0,i}^\top)^\top := \boldsymbol{s}_i \boldsymbol{B}_i \in \mathbb{R}^{d_0 \times (l+1)}, \tag{87}$$

with $\boldsymbol{s}_i = \sigma(\boldsymbol{W}^\top \boldsymbol{x}_i)$. Therefore,

$$
\begin{aligned}
\boldsymbol{h}_i =&\boldsymbol{h}_{i-1} \odot \tilde{\boldsymbol{A}}_i + (\boldsymbol{p}_i \mathbf{1}_{l+1}^\top)\tilde{\boldsymbol{B}}_i \\
=&\boldsymbol{h}_{i-1} \odot \tilde{\boldsymbol{A}}_i + (\boldsymbol{p}_i \mathbf{1}_{l+1}^\top) \odot \boldsymbol{B}_i \\
=&(\boldsymbol{h}_{i-2} \odot \tilde{\boldsymbol{A}}_{i-1} + (\boldsymbol{p}_{i-1} \odot \boldsymbol{s}_i)\boldsymbol{B}_{i-1}) \odot \tilde{\boldsymbol{A}}_i + \boldsymbol{p}_i \boldsymbol{B}_i \\
=&\boldsymbol{h}_{i-2} \odot \tilde{\boldsymbol{A}}_{i-1} \odot \tilde{\boldsymbol{A}}_i + (\boldsymbol{p}_{i-1} \odot \boldsymbol{s}_i)\boldsymbol{B}_{i-1} \odot \tilde{\boldsymbol{A}}_i + (\boldsymbol{p}_i \odot \boldsymbol{s}_i)\boldsymbol{B}_i \\
=&\cdots \\
=&\boldsymbol{h}_0 \odot \tilde{\boldsymbol{A}}_1 \odot \cdots \odot \tilde{\boldsymbol{A}}_i + \sum_{j=1}^{i}(\boldsymbol{p}_j \odot \boldsymbol{s}_j)\boldsymbol{B}_j \odot \tilde{\boldsymbol{A}}_{j+1} \cdots \odot \tilde{\boldsymbol{A}}_i + (\boldsymbol{p}_i \odot \boldsymbol{s}_i)\boldsymbol{B}_i \\
=&\sum_{j=1}^{i}(\boldsymbol{p}_j \odot \boldsymbol{s}_j)\boldsymbol{B}_j \odot (\tilde{\boldsymbol{A}}_i \odot \cdots \odot \tilde{\boldsymbol{A}}_{j+1}) + (\boldsymbol{p}_i \odot \boldsymbol{s}_i)\boldsymbol{B}_i,
\end{aligned}
\tag{88}
$$

Then, given $\boldsymbol{W}_C \in \mathbb{R}^{(l+1) \times d_0}$, we have

$$
\begin{aligned}
\boldsymbol{o}_i =&\boldsymbol{h}_i \boldsymbol{C}_i \\
=&\boldsymbol{h}_i \boldsymbol{W}_C \boldsymbol{p}_i \\
=&\sum_{j=1}^{i}(\boldsymbol{p}_j \odot \boldsymbol{s}_j)\boldsymbol{B}_j(\tilde{\boldsymbol{A}}_i \odot \cdots \odot \tilde{\boldsymbol{A}}_{j+1})\boldsymbol{W}_C \boldsymbol{p}_i + (\boldsymbol{p}_i \odot \boldsymbol{s}_i)\boldsymbol{B}_i \boldsymbol{W}_C \boldsymbol{p}_i \\
=&\sum_{j=1}^{i}(\boldsymbol{G}_{j,i}(\boldsymbol{W}) \odot \boldsymbol{p}_j)\boldsymbol{p}_j^\top \boldsymbol{W}_B^\top \boldsymbol{W}_C \boldsymbol{p}_i,
\end{aligned}
\tag{89}
$$

where the $d_0$-dimensional

$$
\boldsymbol{G}_{j,i}(\boldsymbol{W}) := \begin{cases} (\mathbf{1}_{d_0} - \sigma(\boldsymbol{W}^\top \boldsymbol{p}_{j+1})) \odot \cdots \odot (\mathbf{1}_{d_0} - \sigma(\boldsymbol{W}^\top \boldsymbol{p}_i))\sigma(\boldsymbol{W}^\top \boldsymbol{p}_j), & \text{if } j < i \\ \sigma(\boldsymbol{W}^\top \boldsymbol{p}_i), & \text{if } j = i, \end{cases}
\tag{90}
$$

with $\sigma(\cdot)$ as the sigmoid function. Therefore, we can obtain (3), i.e.,

$$
F(\Psi; \boldsymbol{P}) = \boldsymbol{e}_{d+1}^\top \boldsymbol{o}_{l+1} = \sum_{i=1}^{l+1} G_{i,l+1}(\boldsymbol{w})y_i \boldsymbol{p}_i^\top \boldsymbol{W}_B^\top \boldsymbol{W}_C \boldsymbol{p}_{query},
\tag{91}
$$

where

$$
\begin{aligned}
G_{i,l+1}(\boldsymbol{w}) :=&(\boldsymbol{G}_{i,l+1}(\boldsymbol{W}))_{d+1} \\
=&\begin{cases} \sigma(\boldsymbol{w}^\top \boldsymbol{p}_j) \prod_{k=j+1}^{l+1}(1 - \sigma(\boldsymbol{w}^\top \boldsymbol{p}_k)), & \text{if } j < i \\ \sigma(\boldsymbol{w}^\top \boldsymbol{p}_i), & \text{if } j = i. \end{cases}
\end{aligned}
\tag{92}
$$

$\square$

## F.2. Proof of Lemma 3

*Proof.* (a) When $F(\Psi; \boldsymbol{P}^n) \in (-1, 1)$ for some $n \in [N]$, we have

$$
\frac{\partial \ell(\Psi; \boldsymbol{P}^n, z^n)}{\partial \boldsymbol{W}_C} = -z^n \sum_{i=1}^{l} G_{i,l+1}^n(\boldsymbol{w})y_i^n \boldsymbol{W}_B \boldsymbol{p}_i^n \boldsymbol{p}_{query}^n{}^\top.
\tag{93}
$$

When $t = 0$, we know that with high probability,

$$
|\boldsymbol{w}^{(0)}{}^\top \boldsymbol{x}_j| \lesssim \xi = \frac{1}{d+1},
\tag{94}
$$

$$|\sigma(\boldsymbol{w}^{(0)\top}\boldsymbol{x}_j) - \frac{1}{2}| \lesssim \frac{|1 - e^{\pm\xi}|}{2(1 + e^{\pm\xi})} \lesssim \xi. \tag{95}$$

Then,

$$\frac{1}{2^{l+2-i}}(1 - \xi(l+2-i)) \leq G^{n\,(0)}_{i,l+1}(\boldsymbol{w}) \lesssim \frac{1}{2^{l+2-i}}(1 + \xi(l+2-i)). \tag{96}$$

Let the IDR pattern of $\boldsymbol{\mu}^n_{query}$ be $\boldsymbol{\mu}_j$, $j \in [M_1]$. Note that $\frac{1}{2} \cdot p_a$ fraction of examples correspond to $\boldsymbol{\mu}_j$ with poisoned labels. For different $f$, $y^f_* = 1$ or $-1$ with $1/2$ probability. By Lemma 1, we have for any $i \in l$,

$$\Pr\left(\frac{1}{|\mathcal{B}_b|}\sum_{i\in\mathcal{B}_b}\mathbb{1}[\boldsymbol{x}^n_i \text{ contains } \boldsymbol{\mu}_j \text{ and no } \boldsymbol{v}^*_s] - (1-p_a) \leq -\frac{c}{M_1}(1-p_a)\right) \lesssim e^{-\frac{B(1-p_a)}{M_1}} \leq \epsilon, \tag{97}$$

for some $c \in (0, 1)$ and $\epsilon > 0$ if

$$B \gtrsim (1 - p_a)^{-1}M_1 \log \epsilon^{-1}. \tag{98}$$

By (22), let $\mathcal{B}'_b = \{i : i \in \mathcal{B}_b, \boldsymbol{x}^n_i \text{ contains } \boldsymbol{\mu}_j \text{ and } \boldsymbol{\nu}^*_s, s \in [V]\}$ we have

$$\Pr\left(\Big|\frac{1}{|\mathcal{B}'_b|}\sum_{i\in\mathcal{B}'_b}(\mathbb{1}[y^n_i = z^n] - \mathbb{1}[y^n_i = -z^n])\Big| \geq \sqrt{\frac{\log B}{B}}\right) \leq M_1^{-C}, \tag{99}$$

for some $c \in (0, 1)$ and $C > 1$. Therefore, we have

$$- (\boldsymbol{\mu}^\top_j, \boldsymbol{0}^\top)\eta \cdot \frac{1}{|\mathcal{B}_b|}\sum_{n\in\mathcal{B}_b}\frac{\ell(\Psi^{(0)}; \boldsymbol{P}^n, z^n)}{\partial \boldsymbol{W}^{(0)}_C}(\boldsymbol{\mu}^\top_j, \boldsymbol{0}^\top)^\top$$

$$= (\boldsymbol{\mu}^\top_j, \boldsymbol{0}^\top)\frac{\eta}{|\mathcal{B}_b|}\sum_{n\in\mathcal{B}_b}z^n\sum_{i=1}^l G^n_{i,l+1}(\boldsymbol{w}^{(0)})y^n_i\boldsymbol{W}^{(0)}_B\boldsymbol{p}^n_i\boldsymbol{p}^{n\top}_{query}(\boldsymbol{\mu}^\top_j, \boldsymbol{0}^\top)^\top$$

$$\cdot \mathbb{1}[\boldsymbol{x}^n_i \text{ does not contain any } \boldsymbol{v}^*_s] + (\boldsymbol{\mu}^\top_j, \boldsymbol{0}^\top)\frac{\eta}{|\mathcal{B}_b|}\sum_{n\in\mathcal{B}_b}z^n\sum_{i=1}^l G^n_{i,l+1}(\boldsymbol{w}^{(0)}) \tag{100}$$

$$\cdot y^n_i\boldsymbol{W}^{(0)}_B\boldsymbol{p}^n_i\boldsymbol{p}^{n\top}_{query}(\boldsymbol{\mu}^\top_j, \boldsymbol{0}^\top)^\top\mathbb{1}[\boldsymbol{x}^n_i \text{ contains any } \boldsymbol{v}^*_s]$$

$$\gtrsim \eta \cdot \frac{1}{2M_1}(1-p_a)\sum_{i=1}^l G^n_{i,l+1}(\boldsymbol{w}^{(0)})\beta - \eta \cdot \frac{1}{2M_1}\sum_{i=1}^l G^n_{i,l+1}(\boldsymbol{w}^{(0)})\beta p_a\sqrt{\frac{\log B}{B}}$$

$$\geq \eta\frac{1}{4M_1}(1-p_a)\beta(1 - \xi l),$$

where the last step holds if

$$B \gtrsim (1 - p_a)^{-2}\log \epsilon^{-1}. \tag{101}$$

For $\boldsymbol{\mu}_{j'}$, $j' \neq j$, that does not form a task in the training set, we have

$$-(\boldsymbol{\mu}^\top_{j'}, \boldsymbol{0}^\top)\eta \cdot \frac{1}{|\mathcal{B}_b|}\sum_{n\in\mathcal{B}_b}\frac{\ell(\Psi^{(0)}; \boldsymbol{P}^n, z^n)}{\partial \boldsymbol{W}^{(0)}_C}(\boldsymbol{\mu}^\top_j, \boldsymbol{0}^\top)^\top = 0 \tag{102}$$

For $\boldsymbol{\mu}_{j''}$, $j'' \neq j$, that forms a task in the training set, we have

$$- (\boldsymbol{\mu}^\top_{j''}, \boldsymbol{0}^\top)\eta \cdot \frac{1}{|\mathcal{B}_b|}\sum_{n\in\mathcal{B}_b}\frac{\ell(\Psi^{(0)}; \boldsymbol{P}^n, z^n)}{\partial \boldsymbol{W}^{(0)}_C}(\boldsymbol{\mu}^\top_j, \boldsymbol{0}^\top)^\top$$

$$= (\boldsymbol{\mu}^\top_{j''}, \boldsymbol{0}^\top)\frac{\eta}{|\mathcal{B}_b|}\sum_{n\in\mathcal{B}_b}z^n\sum_{i=1}^l G^n_{i,l+1}(\boldsymbol{w}^{(0)})y^n_i\boldsymbol{W}^{(0)}_B\boldsymbol{p}^n_i\boldsymbol{p}^{n\top}_{query}(\boldsymbol{\mu}^\top_j, \boldsymbol{0}^\top)^\top \tag{103}$$

$$\lesssim -\eta \cdot \frac{1}{4M_1}(1-p_a)\beta(1 - \xi l).$$

For $\boldsymbol{\nu}_k, \boldsymbol{\nu}_{k'}$ with $k, k' \in [M_2]$, we have

$$\left| -(\boldsymbol{\nu}_k^\top, 0^\top)\eta \cdot \frac{1}{|\mathcal{B}_b|} \sum_{n \in \mathcal{B}_b} \frac{\ell(\Psi^{(0)}; \boldsymbol{P}^n, z^n)}{\partial \boldsymbol{W}_C^{(0)}} (\boldsymbol{\mu}_j^\top, 0^\top)^\top \right| \leq \frac{\eta\beta}{M_1 M_2} \sqrt{\frac{\log B}{B}}, \tag{104}$$

$$\left| -(\boldsymbol{\mu}_j^\top, 0^\top)\eta \cdot \frac{1}{|\mathcal{B}_b|} \sum_{n \in \mathcal{B}_b} \frac{\ell(\Psi^{(0)}; \boldsymbol{P}^n, z^n)}{\partial \boldsymbol{W}_C^{(0)}} (\boldsymbol{\nu}_k^\top, 0^\top)^\top \right| \leq \frac{\eta\beta}{M_2 M_1} \sqrt{\frac{\log B}{B}}. \tag{105}$$

$$\left| -(\boldsymbol{\nu}_{k'}^\top, 0^\top)\eta \cdot \frac{1}{|\mathcal{B}_b|} \sum_{n \in \mathcal{B}_b} \frac{\ell(\Psi^{(0)}; \boldsymbol{P}^n, z^n)}{\partial \boldsymbol{W}_C^{(0)}} (\boldsymbol{\nu}_k^\top, 0^\top)^\top \right| \leq \frac{\eta\beta}{M_2^2} \sqrt{\frac{\log B}{B}}. \tag{106}$$

$$\left| -(\boldsymbol{\nu}_k^\top, 0^\top)\eta \cdot \frac{1}{|\mathcal{B}_b|} \sum_{n \in \mathcal{B}_b} \frac{\ell(\Psi^{(0)}; \boldsymbol{P}^n, z^n)}{\partial \boldsymbol{W}_C^{(0)}} (\boldsymbol{\nu}_k^\top, 0^\top)^\top \right| \leq \frac{\eta\beta}{M_2} \sqrt{\frac{\log B}{B}}. \tag{107}$$

Since that for $\boldsymbol{x}_i^n$ that contains $\boldsymbol{\nu}_s^*$ for a certain $s \in [V]$,

$$\Pr(y_i^n = z^n) = \Pr(y_i^n = -z^n) = \frac{1}{2}, \tag{108}$$

we have

$$\left| (\boldsymbol{\nu}_s^{*\top}, 0^\top)\eta \cdot \frac{1}{|\mathcal{B}_b|} \sum_{n \in \mathcal{B}_b} \frac{\ell(\Psi^{(0)}; \boldsymbol{P}^n, z^n)}{\partial \boldsymbol{W}_C^{(0)}} (\boldsymbol{\mu}_j^\top, 0^\top)^\top \right|$$

$$= \left| (\boldsymbol{\nu}_s^{*\top}, 0^\top) \frac{\eta}{|\mathcal{B}_b|} \sum_{n \in \mathcal{B}_b} z^n \sum_{i=1}^l G_{i,l+1}^n(\boldsymbol{w}^{(0)}) y_i^n \boldsymbol{W}_B^{(0)} \boldsymbol{p}_i^n \boldsymbol{p}_{query}^n{}^\top (\boldsymbol{\mu}_j^\top, 0^\top)^\top \right| \tag{109}$$

$$\leq \frac{\eta\beta p_a \kappa_*}{M_1 V} \cdot \sqrt{\frac{\log B}{B}},$$

Suppose that the conclusion holds when $t = t_0$. Then, when $t = t_0 + 1$, we have

$$-(\boldsymbol{\mu}_j^\top, 0^\top)\eta \cdot \sum_{b=1}^{t_0+1} \frac{1}{|\mathcal{B}_b|} \sum_{n \in \mathcal{B}_b} \frac{\ell(\Psi^{(b)}; \boldsymbol{P}^n, z^n)}{\partial \boldsymbol{W}_C^{(b)}} (\boldsymbol{\mu}_j^\top, 0^\top)^\top$$

$$= (\boldsymbol{\mu}_j^\top, 0^\top) \sum_{b=1}^{t_0+1} \frac{\eta}{|\mathcal{B}_b|} \sum_{n \in \mathcal{B}_b} z^n \sum_{i=1}^l G_{i,l+1}^n(\boldsymbol{w}^{(b)}) y_i^n \boldsymbol{W}_B^{(b)} \boldsymbol{p}_i^n \boldsymbol{p}_{query}^n{}^\top (\boldsymbol{\mu}_j^\top, 0^\top)^\top \tag{110}$$

$$\gtrsim \eta \cdot \sum_{b=1}^{t_0+1} \frac{1}{2M_1} (1 - p_a) \sum_{i=1}^l G_{i,l+1}^n(\boldsymbol{w}^{(t_0)})\beta$$

$$\gtrsim \eta(t_0 + 1) \frac{1}{M_1} (1 - p_a)\beta.$$

The last step holds since $\sum_{i=1}^l G_{i,l+1}^n(\boldsymbol{w}^{(t_0)}) \gtrsim 1$. Similarly, we have that for any $s \in [V]$,

$$\left| (\boldsymbol{\nu}_s^{*\top}, 0^\top)\eta \cdot \sum_{b=1}^{t_0+1} \frac{1}{|\mathcal{B}_b|} \sum_{n \in \mathcal{B}_b} \frac{\ell(\Psi^{(b)}; \boldsymbol{P}^n, z^n)}{\partial \boldsymbol{W}_C^{(b)}} (\boldsymbol{\mu}_j^\top, 0^\top)^\top \right| \leq \frac{\eta\beta(t_0 + 1)p_a \kappa_*}{M_1} \cdot \sqrt{\frac{\log B}{B}}, \tag{111}$$

For $\boldsymbol{\mu}_{j'}, j' \neq j$, that forms a task in the training set, we have

$$-(\boldsymbol{\mu}_{j'}^\top, 0^\top)\eta \cdot \sum_{b=1}^{t_0+1} \frac{1}{|\mathcal{B}_b|} \sum_{n \in \mathcal{B}_b} \frac{\ell(\Psi^{(b)}; \boldsymbol{P}^n, z^n)}{\partial \boldsymbol{W}_C^{(b)}} (\boldsymbol{\mu}_j^\top, 0^\top)^\top = 0 \tag{112}$$

For $\boldsymbol{\mu}_{j''}, j'' \neq j$, that forms a task in the training set, we have

$$
\begin{aligned}
&- (\boldsymbol{\mu}_{j''}^{\top}, 0^{\top})\eta \cdot \sum_{b=1}^{t_0+1} \frac{1}{|\mathcal{B}_b|} \sum_{n \in \mathcal{B}_b} \frac{\ell(\Psi^{(b)}; \boldsymbol{P}^n, z^n)}{\partial \boldsymbol{W}_C^{(b)}} (\boldsymbol{\mu}_j^{\top}, 0^{\top})^{\top} \\
&\leq (\boldsymbol{\mu}_j^{\top}, 0^{\top})\eta \cdot \sum_{b=1}^{t_0+1} \frac{1}{|\mathcal{B}_b|} \sum_{n \in \mathcal{B}_b} \frac{\ell(\Psi^{(b)}; \boldsymbol{P}^n, z^n)}{\partial \boldsymbol{W}_C^{(b)}} (\boldsymbol{\mu}_j^{\top}, 0^{\top})^{\top}.
\end{aligned}
\tag{113}
$$

For $\boldsymbol{\nu}_k, \boldsymbol{\nu}_{k'}$ with $k \neq k' \in [M_2]$, we have

$$
\left| - (\boldsymbol{\nu}_k^{\top}, 0^{\top})\eta \cdot \sum_{b=1}^{t_0+1} \frac{1}{|\mathcal{B}_b|} \sum_{n \in \mathcal{B}_b} \frac{\ell(\Psi^{(b)}; \boldsymbol{P}^n, z^n)}{\partial \boldsymbol{W}_C^{(b)}} (\boldsymbol{\mu}_j^{\top}, 0^{\top})^{\top} \right| \leq \frac{\eta(t_0+1)\beta}{M_1 M_2} \sqrt{\frac{\log B}{B}},
\tag{114}
$$

$$
\left| - (\boldsymbol{\mu}_j^{\top}, 0^{\top})\eta \cdot \sum_{b=1}^{t_0+1} \frac{1}{|\mathcal{B}_b|} \sum_{n \in \mathcal{B}_b} \frac{\ell(\Psi^{(b)}; \boldsymbol{P}^n, z^n)}{\partial \boldsymbol{W}_C^{(b)}} (\boldsymbol{\nu}_k^{\top}, 0^{\top})^{\top} \right| \leq \frac{\eta(t_0+1)\beta}{M_1 M_2} \sqrt{\frac{\log B}{B}},
\tag{115}
$$

$$
\left| - (\boldsymbol{\nu}_k^{\top}, 0^{\top})\eta \cdot \sum_{b=1}^{t_0+1} \frac{1}{|\mathcal{B}_b|} \sum_{n \in \mathcal{B}_b} \frac{\ell(\Psi^{(b)}; \boldsymbol{P}^n, z^n)}{\partial \boldsymbol{W}_C^{(b)}} (\boldsymbol{\nu}_k^{\top}, 0^{\top})^{\top} \right| \leq \frac{\eta(t_0+1)\beta}{M_2} \sqrt{\frac{\log B}{B}},
\tag{116}
$$

$$
\left| - (\boldsymbol{\nu}_{k'}^{\top}, 0^{\top})\eta \cdot \sum_{b=1}^{t_0+1} \frac{1}{|\mathcal{B}_b|} \sum_{n \in \mathcal{B}_b} \frac{\ell(\Psi^{(b)}; \boldsymbol{P}^n, z^n)}{\partial \boldsymbol{W}_C^{(b)}} (\boldsymbol{\nu}_k^{\top}, 0^{\top})^{\top} \right| \leq \frac{\eta(t_0+1)\beta}{M_2^2} \sqrt{\frac{\log B}{B}},
\tag{117}
$$

Then, we complete the induction.

(b) We then characterize the gradient updates of $\boldsymbol{W}_B$. We have that when $F(\Psi; \boldsymbol{P}^n) \in (-1, 1)$ for some $n \in [N]$,

$$
\frac{\partial \ell(\Psi; \boldsymbol{P}^n, z^n)}{\partial \boldsymbol{W}_B} = - z^n \sum_{i=1}^{l+1} G_{i,l+1}^n(\boldsymbol{w}) y_i \boldsymbol{W}_C \boldsymbol{p}_{query} \boldsymbol{p}_i^{\top}.
\tag{118}
$$

We also use induction to complete the proof. Similar to the analysis of $\boldsymbol{W}_C$, we have that when $t = 0$,

$$
\begin{aligned}
&- (\boldsymbol{\mu}_j^{\top}, 0^{\top})\eta \cdot \frac{1}{|\mathcal{B}_b|} \sum_{n \in \mathcal{B}_b} \frac{\ell(\Psi^{(0)}; \boldsymbol{P}^n, z^n)}{\partial \boldsymbol{W}_B^{(0)}} (\boldsymbol{\mu}_j^{\top}, 0^{\top})^{\top} \\
&= (\boldsymbol{\mu}_j^{\top}, 0^{\top}) \frac{\eta}{|\mathcal{B}_b|} \sum_{n \in \mathcal{B}_b} z^n \sum_{i=1}^{l} G_{i,l+1}^n(\boldsymbol{w}^{(0)}) y_i^n \boldsymbol{W}_C^{(0)} \boldsymbol{p}_{query}^n \boldsymbol{p}_i^{n\top} (\boldsymbol{\mu}_j^{\top}, 0^{\top})^{\top} \\
&\gtrsim \eta \cdot \frac{1}{2M_1}(1 - p_a) \sum_{i=1}^{l} G_{i,l+1}^n(\boldsymbol{w}^{(0)})\beta - \eta \cdot \frac{1}{2M_1} \sum_{i=1}^{l} G_{i,l+1}^n(\boldsymbol{w}^{(0)})\beta p_a \sqrt{\frac{\log B}{B}} \\
&\geq \eta \frac{1}{4M_1}(1 - p_a)\beta(1 - \xi l).
\end{aligned}
\tag{119}
$$

For $\boldsymbol{\mu}_{j'}, j' \neq j$, that does not form a task in the training stage, we have

$$
- (\boldsymbol{\mu}_{j'}^{\top}, 0^{\top})\eta \cdot \frac{1}{|\mathcal{B}_b|} \sum_{n \in \mathcal{B}_b} \frac{\ell(\Psi^{(0)}; \boldsymbol{P}^n, z^n)}{\partial \boldsymbol{W}_B^{(0)}} (\boldsymbol{\mu}_j^{\top}, 0^{\top})^{\top} = 0.
\tag{120}
$$

For $\boldsymbol{\mu}_{j''}, j'' \neq j$, that forms a task in the training stage, we have

$$
- (\boldsymbol{\mu}_{j''}^{\top}, 0^{\top})\eta \cdot \frac{1}{|\mathcal{B}_b|} \sum_{n \in \mathcal{B}_b} \frac{\ell(\Psi^{(0)}; \boldsymbol{P}^n, z^n)}{\partial \boldsymbol{W}_B^{(0)}} (\boldsymbol{\mu}_j^{\top}, 0^{\top})^{\top} \leq -\eta \cdot \frac{1}{4M_1}(1 - p_a)\beta(1 - \xi l).
\tag{121}
$$

For $\boldsymbol{\nu}_k, \boldsymbol{\nu}_{k'}$ with $k \neq k' \in [M_2]$, we have

$$\left| -(\boldsymbol{\nu}_k{}^\top, 0^\top)\eta \cdot \frac{1}{|\mathcal{B}_b|} \sum_{n \in \mathcal{B}_b} \frac{\ell(\Psi^{(0)}; \boldsymbol{P}^n, z^n)}{\partial \boldsymbol{W}_B^{(0)}} (\boldsymbol{\mu}_j^\top, 0^\top)^\top \right| \leq \frac{\eta\beta}{M_1 M_2} \sqrt{\frac{\log B}{B}}, \tag{122}$$

$$\left| -(\boldsymbol{\mu}_j{}^\top, 0^\top)\eta \cdot \frac{1}{|\mathcal{B}_b|} \sum_{n \in \mathcal{B}_b} \frac{\ell(\Psi^{(0)}; \boldsymbol{P}^n, z^n)}{\partial \boldsymbol{W}_B^{(0)}} (\boldsymbol{\nu}_k^\top, 0^\top)^\top \right| \leq \frac{\eta\beta}{M_1 M_2} \sqrt{\frac{\log B}{B}}. \tag{123}$$

$$\left| -(\boldsymbol{\nu}_k{}^\top, 0^\top)\eta \cdot \frac{1}{|\mathcal{B}_b|} \sum_{n \in \mathcal{B}_b} \frac{\ell(\Psi^{(0)}; \boldsymbol{P}^n, z^n)}{\partial \boldsymbol{W}_B^{(0)}} (\boldsymbol{\nu}_k^\top, 0^\top)^\top \right| \leq \frac{\eta\beta}{M_2} \sqrt{\frac{\log B}{B}}. \tag{124}$$

$$\left| -(\boldsymbol{\nu}_{k'}{}^\top, 0^\top)\eta \cdot \frac{1}{|\mathcal{B}_b|} \sum_{n \in \mathcal{B}_b} \frac{\ell(\Psi^{(0)}; \boldsymbol{P}^n, z^n)}{\partial \boldsymbol{W}_B^{(0)}} (\boldsymbol{\nu}_k^\top, 0^\top)^\top \right| \leq \frac{\eta\beta}{M_2^2} \sqrt{\frac{\log B}{B}}. \tag{125}$$

We also have that for any $s \in [V]$,

$$\left| (\boldsymbol{\nu}_s^{*\top}, 0^\top)\eta \cdot \frac{1}{|\mathcal{B}_b|} \sum_{n \in \mathcal{B}_b} \frac{\ell(\Psi^{(0)}; \boldsymbol{P}^n, z^n)}{\partial \boldsymbol{W}_B^{(0)}} (\boldsymbol{\mu}_j^\top, 0^\top)^\top \right| \leq \frac{\eta\beta p_a \kappa_*}{M_1 V} \cdot \sqrt{\frac{\log B}{B}}, \tag{126}$$

Therefore, the conclusions hold when $t = 0$. Suppose that the conclusions also hold when $t = t_0$. Then, when $t = t_0 + 1$, we have

$$-(\boldsymbol{\mu}_j^\top, 0^\top)\eta \cdot \sum_{b=1}^{t_0+1} \frac{1}{|\mathcal{B}_b|} \sum_{n \in \mathcal{B}_b} \frac{\ell(\Psi^{(b)}; \boldsymbol{P}^n, z^n)}{\partial \boldsymbol{W}_B^{(b)}} (\boldsymbol{\mu}_j^\top, 0^\top)^\top$$
$$\gtrsim \eta \cdot \sum_{c=1}^{t_0+1} \frac{1}{2M_1} (1 - p_a) \sum_{i=1}^{l} G_{i,l+1}^n(\boldsymbol{w}^{(t_0)})\beta \tag{127}$$
$$\gtrsim \eta(t_0 + 1) \frac{1}{M_1}(1 - p_a)\beta.$$

For $\boldsymbol{\mu}_{j'}, j' \neq j$, that does not form a task in the training set, we have

$$-(\boldsymbol{\mu}_{j'}{}^\top, 0^\top)\eta \cdot \sum_{b=1}^{t_0+1} \frac{1}{|\mathcal{B}_b|} \sum_{n \in \mathcal{B}_b} \frac{\ell(\Psi^{(b)}; \boldsymbol{P}^n, z^n)}{\partial \boldsymbol{W}_B^{(b)}} (\boldsymbol{\mu}_j^\top, 0^\top)^\top = 0 \tag{128}$$

For $\boldsymbol{\mu}_{j''}, j'' \neq j$, that forms a task in the training set, we have

$$-(\boldsymbol{\mu}_{j''}{}^\top, 0^\top)\eta \cdot \sum_{b=1}^{t_0+1} \frac{1}{|\mathcal{B}_b|} \sum_{n \in \mathcal{B}_b} \frac{\ell(\Psi^{(b)}; \boldsymbol{P}^n, z^n)}{\partial \boldsymbol{W}_B^{(b)}} (\boldsymbol{\mu}_j^\top, 0^\top)^\top$$
$$\leq -\eta(t_0 + 1) \frac{1}{M_1}(1 - p_a)\beta. \tag{129}$$

For $\boldsymbol{\nu}_k, \boldsymbol{\nu}_{k'}$ with $k \neq k' \in [M_2]$, we have

$$\left| -(\boldsymbol{\nu}_k{}^\top, 0^\top)\eta \cdot \sum_{b=1}^{t_0+1} \frac{1}{|\mathcal{B}_b|} \sum_{n \in \mathcal{B}_b} \frac{\ell(\Psi^{(b)}; \boldsymbol{P}^n, z^n)}{\partial \boldsymbol{W}_B^{(b)}} (\boldsymbol{\mu}_j^\top, 0^\top)^\top \right| \leq \frac{\eta(t_0 + 1)\beta}{M_1 M_2} \sqrt{\frac{\log B}{B}}, \tag{130}$$

$$\left| -(\boldsymbol{\mu}_j{}^\top, 0^\top)\eta \cdot \sum_{b=1}^{t_0+1} \frac{1}{|\mathcal{B}_b|} \sum_{n \in \mathcal{B}_b} \frac{\ell(\Psi^{(b)}; \boldsymbol{P}^n, z^n)}{\partial \boldsymbol{W}_B^{(b)}} (\boldsymbol{\nu}_k^\top, 0^\top)^\top \right| \leq \frac{\eta(t_0 + 1)\beta}{M_1 M_2} \sqrt{\frac{\log B}{B}}. \tag{131}$$

$$\left| -(\boldsymbol{\nu}_k{}^\top, 0^\top)\eta \cdot \sum_{b=1}^{t_0+1} \frac{1}{|\mathcal{B}_b|} \sum_{n \in \mathcal{B}_b} \frac{\ell(\Psi^{(b)}; \boldsymbol{P}^n, z^n)}{\partial \boldsymbol{W}_B^{(b)}} (\boldsymbol{\nu}_k^\top, 0^\top)^\top \right| \leq \frac{\eta(t_0 + 1)\beta}{M_2} \sqrt{\frac{\log B}{B}}. \tag{132}$$

$$\left| -(\boldsymbol{\nu}_{k'}{}^\top, 0^\top)\eta \cdot \sum_{b=1}^{t_0+1} \frac{1}{|\mathcal{B}_b|} \sum_{n\in\mathcal{B}_b} \frac{\ell(\Psi^{(b)}; \boldsymbol{P}^n, z^n)}{\partial \boldsymbol{W}_B^{(b)}}(\boldsymbol{\nu}_k^\top, 0^\top)^\top \right| \le \frac{\eta(t_0+1)\beta}{M_2^2}\sqrt{\frac{\log B}{B}}. \tag{133}$$

We also have that for any $s \in [V]$,

$$\left| (\boldsymbol{\nu}_s^{*\top}, 0^\top)\eta \cdot \sum_{b=1}^{t_0+1} \frac{1}{|\mathcal{B}_b|} \sum_{n\in\mathcal{B}_b} \frac{\ell(\Psi^{(b)}; \boldsymbol{P}^n, z^n)}{\partial \boldsymbol{W}_B^{(b)}}(\boldsymbol{\mu}_j^\top, 0^\top)^\top \right| \le \frac{\eta\beta(t_0+1)p_a\kappa_*}{M_1 V} \cdot \sqrt{\frac{\log B}{B}}, \tag{134}$$

$\square$

### F.3. Proof of Lemma 4

*Proof.* When $F(\Psi; \boldsymbol{P}^n) \in (-1, 1)$ for some $n \in [N]$,

$$
\begin{aligned}
&\frac{\partial \ell(\Psi; \boldsymbol{P}^n, z^n)}{\partial \boldsymbol{w}} \\
=& -z^n \sum_{i=1}^l y_i^n \boldsymbol{p}_i^{n\top} \boldsymbol{W}_B^\top \boldsymbol{W}_C \boldsymbol{p}_{query}^n \frac{\partial G_{i,l+1}^n(\boldsymbol{w})}{\partial \boldsymbol{w}} \\
=& -z^n \sum_{i=1}^l y_i^n \boldsymbol{p}_i^{n\top} \boldsymbol{W}_B^\top \boldsymbol{W}_C \boldsymbol{p}_{query}^n \frac{\partial \prod_{j=i+1}^{l+1}(1-\sigma(\boldsymbol{w}^\top \boldsymbol{p}_j^n))\sigma(\boldsymbol{w}^\top \boldsymbol{p}_i^n)}{\partial \boldsymbol{w}} \\
=& -z^n \sum_{i=1}^l y_i^n \boldsymbol{p}_i^{n\top} \boldsymbol{W}_B^\top \boldsymbol{W}_C \boldsymbol{p}_{query}^n \Big( \sum_{s=i+1}^{l+1} \prod_{j=i+1, j\neq s}^{l+1} (1-\sigma(\boldsymbol{w}^\top \boldsymbol{p}_j^n)\mathbb{1}[j<l+1])\sigma(\boldsymbol{w}^\top \boldsymbol{p}_i^n) \\
& \cdot \frac{\partial(1-\sigma(\boldsymbol{w}^\top \boldsymbol{p}_s^n))}{\partial \boldsymbol{w}} + \prod_{j=i+1}^{l+1}(1-\sigma(\boldsymbol{w}^\top \boldsymbol{p}_j^n))\frac{\partial\sigma(\boldsymbol{w}^\top \boldsymbol{p}_i^n)}{\partial \boldsymbol{w}}\Big) \\
=& -z^n \sum_{i=1}^l y_i^n \boldsymbol{p}_i^{n\top} \boldsymbol{W}_B^\top \boldsymbol{W}_C \boldsymbol{p}_{query}^n \Big( \sum_{s=i+1}^{l+1} \prod_{j=i+1, j\neq s}^{l+1} (1-\sigma(\boldsymbol{w}^\top \boldsymbol{p}_j^n)\mathbb{1}[j<l+1])\sigma(\boldsymbol{w}^\top \boldsymbol{p}_i^n) \\
& \cdot (1-\sigma(\boldsymbol{w}^\top \boldsymbol{p}_s^n))\sigma(\boldsymbol{w}^\top \boldsymbol{p}_s^n)(-\boldsymbol{p}_s^n) + \prod_{j=i+1}^{l+1}(1-\sigma(\boldsymbol{w}^\top \boldsymbol{p}_j^n))(1-\sigma(\boldsymbol{w}^\top \boldsymbol{p}_i^n))\sigma(\boldsymbol{w}^\top \boldsymbol{p}_i^n)\boldsymbol{p}_i^n \\
=& z^n \sum_{i=1}^l y_i^n \boldsymbol{p}_i^{n\top} \boldsymbol{W}_B^\top \boldsymbol{W}_C \boldsymbol{p}_{query}^n \Big( \sum_{s=i+1}^{l+1} \prod_{j=i+1}^{l+1} (1-\sigma(\boldsymbol{w}^\top \boldsymbol{p}_j^n)\mathbb{1}[j<l+1]) \cdot \sigma(\boldsymbol{w}^\top \boldsymbol{p}_s^n) \\
& \cdot \sigma(\boldsymbol{w}^\top \boldsymbol{p}_i^n)\boldsymbol{p}_s^n - \prod_{j=i}^{l+1}(1-\sigma(\boldsymbol{w}^\top \boldsymbol{p}_j^n))\sigma(\boldsymbol{w}^\top \boldsymbol{p}_i^n)\boldsymbol{p}_i^n\Big) \\
=& z^n \sum_{i=1}^l y_i^n \boldsymbol{p}_i^{n\top} \boldsymbol{W}_B^\top \boldsymbol{W}_C \boldsymbol{p}_{query}^n G_{i,l+1}^n(\boldsymbol{w})\Big( \sum_{s=i+1}^{l+1} \sigma(\boldsymbol{w}^\top \boldsymbol{p}_s^n)\boldsymbol{p}_s^n - (1-\sigma(\boldsymbol{w}^\top \boldsymbol{p}_i^n))\boldsymbol{p}_i^n\Big).
\end{aligned}
\tag{135}
$$

When $t = 1$, we have

$$
\begin{aligned}
\boldsymbol{w}^{(1)} =& \boldsymbol{w}^{(0)} - \frac{\eta}{|\mathcal{B}_1|} \sum_{n\in\mathcal{B}_1} \frac{\partial \ell(\Psi; \boldsymbol{P}^n, z^n)}{\partial \boldsymbol{w}^{(0)}} \\
=& \boldsymbol{w}^{(0)} - \frac{\eta}{|\mathcal{B}_1|} \sum_{n\in\mathcal{B}_1} z^n \sum_{i=1}^l y_i^n \boldsymbol{p}_i^{n\top} \boldsymbol{W}_B^{(0)\top} \boldsymbol{W}_C^{(0)} \boldsymbol{p}_{query}^n G_{i,l+1}^n(\boldsymbol{w}^{(0)}) \\
& \cdot \Big( \sum_{s=i+1}^{l+1} \sigma(\boldsymbol{w}^{(0)\top} \boldsymbol{p}_s^n)\boldsymbol{p}_s^n - (1-\sigma(\boldsymbol{w}^{(0)\top} \boldsymbol{p}_i^n))\boldsymbol{p}_i^n\Big)
\end{aligned}
\tag{136}
$$

For $\boldsymbol{p}_i^n$ that contains a $\boldsymbol{v}_s^*$, the corresponding $y_i^n$ is consistent with $z^n$ with a probability of $1/2$. Given Hoeffding's bound (22), this part generates a gradient update as

$$
\left\| \frac{\eta}{|\mathcal{B}_1|} \sum_{n \in \mathcal{B}_1} z^n \sum_{1 \leq i \leq l, \boldsymbol{p}_i^n \text{ does not contain any } \boldsymbol{v}_s^*} y_i^n \boldsymbol{p}_i^{n\top} \boldsymbol{W}_B^{(0)\top} \boldsymbol{W}_C^{(0)} \boldsymbol{p}_{query}^n G_{i,l+1}^n(\boldsymbol{w}^{(0)}) \right.
$$
$$
\left. \cdot \left( \sum_{s=i+1}^{l+1} \sigma(\boldsymbol{w}^{(0)\top} \boldsymbol{p}_s^n) \boldsymbol{p}_s^n - (1 - \sigma(\boldsymbol{w}^{(0)\top} \boldsymbol{p}_i^n)) \boldsymbol{p}_i^n \right) \right\| \tag{137}
$$
$$
\leq \eta \sqrt{\frac{\log B}{B}}
$$

by (96) and $\sum_{i=1}^l \frac{l}{2^l} \leq 2$. Then, with a high probability, for $s \in [V]$, $\xi = 1/(d+1)$,

$$
\boldsymbol{v}_s^{*\top} \boldsymbol{w}^{(1)}
$$
$$
\leq \xi + \eta \sqrt{\frac{\log B}{B}} - \eta \beta^2 \frac{1}{|\mathcal{B}_1|} \sum_{n \in \mathcal{B}_b} \sum_{1 \leq i \leq l, \boldsymbol{p}_i^n \text{ does not contain any } \boldsymbol{v}_s^*}^l G_{i,l+1}^n(\boldsymbol{w}^{(0)})
$$
$$
\cdot \left( \sum_{s=i+1}^{l+1} \sigma(\boldsymbol{w}^{(0)\top} \boldsymbol{p}_s^n) \boldsymbol{v}_s^{*\top} \boldsymbol{p}_s^n - (1 - \sigma(\boldsymbol{w}^{(0)\top} \boldsymbol{p}_i^n)) \boldsymbol{v}_s^{*\top} \boldsymbol{p}_i^n \right)
$$
$$
\lesssim \xi + \eta \sqrt{\frac{\log B}{B}} - \eta \beta^2 \sum_{i=1}^l \frac{1}{2^{l+2-i}V} \cdot \kappa_a \sum_{s=i+1}^{l+1} \frac{1}{2}(1 - p_a) \tag{138}
$$
$$
= \xi + \eta \sqrt{\frac{\log B}{B}} - \eta \beta^2 \sum_{i=1}^l \frac{\kappa_a}{2^{l+2-i}V} \cdot \frac{(1 - p_a)(l - i + 1)}{2}
$$
$$
= \xi + \eta \sqrt{\frac{\log B}{B}} - \eta \beta^2 \cdot \sum_{i=1}^l \frac{\kappa_a i}{2^{2+i}V} \cdot \frac{1 - p_a}{2}
$$
$$
\lesssim \xi + \eta \sqrt{\frac{\log B}{B}} - \frac{\eta \beta^2 \kappa_a (1 - p_a)}{V}
$$
$$
\lesssim - \frac{\eta \beta^2 \kappa_a (1 - p_a)}{V}.
$$

The second step comes from (96) and the fact that

$$
\Pr \left( \left| \frac{1}{l|\mathcal{B}_1|} \sum_{n \in \mathcal{B}_1} \sum_{i=1}^l \mathbb{1}[\boldsymbol{p}_i^n \text{ does not contain any } \boldsymbol{v}_s^*] G_{i,l+1}^n(\boldsymbol{w}^{(0)}) \sum_{s=i+1}^{l+1} \sigma(\boldsymbol{w}^{(0)\top} \boldsymbol{p}_s^n) \right. \right.
$$
$$
\left. \cdot \boldsymbol{v}_s^{*\top} \boldsymbol{p}_s^n - (1 - p_a) \mathbb{E}[\frac{1}{l|\mathcal{B}_1|} \sum_{n \in \mathcal{B}_1} \sum_{i=1}^l G_{i,l+1}^n(\boldsymbol{w}^{(0)}) \sum_{s=i+1}^{l+1} \sigma(\boldsymbol{w}^{(0)\top} \boldsymbol{p}_s^n) \boldsymbol{v}_s^{*\top} \boldsymbol{p}_s^n] \right|
$$
$$
\geq c \cdot (1 - p_a) \mathbb{E}[\frac{1}{l|\mathcal{B}_1|} \sum_{n \in \mathcal{B}_1} \sum_{i=1}^l G_{i,l+1}^n(\boldsymbol{w}^{(0)}) \sum_{s=i+1}^{l+1} \sigma(\boldsymbol{w}^{(0)\top} \boldsymbol{p}_s^n) \boldsymbol{v}_s^{*\top} \boldsymbol{p}_s^n] \right) \tag{139}
$$
$$
\lesssim e^{-lB(1-p_a)^2 c^2}
$$
$$
\leq \epsilon
$$

for some $c \in (0, 1)$, and

$$
Bl \geq (1 - p_a)^{-2} \log \epsilon^{-1} \tag{140}
$$

by Lemma 2 since $\boldsymbol{p}_i^n$ contains $\boldsymbol{v}_s^*$ with a probability of $p_a/V$. The last step holds with a high probability if

$$
B \gtrsim \beta^{-4} \kappa_a^{-2} (1 - p_a)^{-2} V^2 \log \epsilon^{-1}. \tag{141}
$$

We can also derive that for any $j \in [M_1]$,

$$
(\boldsymbol{\mu}_j^\top, 0^\top)\boldsymbol{w}^{(1)}
$$

$$
\leq \xi + \frac{\eta}{M_1}\sqrt{\frac{\log B}{B}} - \frac{\eta\beta^2}{|\mathcal{B}_1|}\sum_{n \in \mathcal{B}_b}\sum_{1 \leq i \leq l, \boldsymbol{p}_i^n \text{ does not contain any } \boldsymbol{v}_s^*}^{l} G_{i,l+1}^n(\boldsymbol{w}^{(0)})(\sum_{s=i+1}^{l+1}\sigma(\boldsymbol{w}^{(0)\top}\boldsymbol{p}_s^n)
$$

$$
\cdot (\boldsymbol{\mu}_j^\top, 0^\top)\boldsymbol{p}_s^n - (1 - \sigma(\boldsymbol{w}^{(0)\top}\boldsymbol{p}_i^n))(\boldsymbol{\mu}_j^\top, 0^\top)\boldsymbol{p}_i^n)
$$

$$
\lesssim \xi + \frac{\eta}{M_1}\sqrt{\frac{\log B}{B}} - \eta\beta^2\sum_{i=1}^{l}\frac{1}{2^{l+2-i}} \cdot \frac{(1-p_a)}{2M_1}(l - i + 1) \tag{142}
$$

$$
\lesssim \xi + \frac{\eta}{M_1}\sqrt{\frac{\log B}{B}} - \frac{\eta(1-p_a)\beta^2}{M_1}
$$

$$
\lesssim - \frac{\eta(1-p_a)\beta^2}{M_1}.
$$

The second step of (142) comes from the fact that

$$
\Pr\Big(\Big|\frac{1}{l|\mathcal{B}_1|}\sum_{n \in \mathcal{B}_1}\sum_{i=1}^{l}\mathbb{1}[\boldsymbol{p}_i^n \text{ does not contain any } \boldsymbol{v}_s^*]G_{i,l+1}^n(\boldsymbol{w}^{(0)})\sum_{s=i+1}^{l+1}\sigma(\boldsymbol{w}^{(0)\top}\boldsymbol{p}_s^n)
$$

$$
- (1-p_a)\mathbb{E}[\frac{1}{l|\mathcal{B}_1|}\sum_{n \in \mathcal{B}_1}\sum_{i=1}^{l}G_{i,l+1}^n(\boldsymbol{w}^{(0)})\sum_{s=i+1}^{l+1}\sigma(\boldsymbol{w}^{(0)\top}\boldsymbol{p}_s^n)]\Big|
$$

$$
\geq c \cdot (1-p_a)\mathbb{E}[\frac{1}{l|\mathcal{B}_1|}\sum_{n \in \mathcal{B}_1}\sum_{i=1}^{l}G_{i,l+1}^n(\boldsymbol{w}^{(0)})\sum_{s=i+1}^{l+1}\sigma(\boldsymbol{w}^{(0)\top}\boldsymbol{p}_s^n)]\Big) \tag{143}
$$

$$
\lesssim e^{-lB(1-p_a)^2c^2}
$$

$$
\leq M_1^{-C}
$$

for some $c \in (0, 1), C > 1$, and

$$
Bl \geq (1-p_a)^{-2}\log \epsilon^{-1} \tag{144}
$$

by Lemma 2 since $\boldsymbol{p}_i^n$ does not contain any $\boldsymbol{v}_s^*$ with a probability of $1 - p_a$.

The last step of (142) holds if $B \gtrsim \beta^{-4}$ and $\xi \lesssim \frac{1}{M_1}$. Similarly, we also have

$$
(\boldsymbol{\mu}_j^\top, 0^\top)\boldsymbol{w}^{(1)}
$$

$$
\geq -\xi - \frac{\eta}{M_1}\sqrt{\frac{\log B}{B}} - \frac{\eta\beta^2}{|\mathcal{B}_1|}\sum_{n \in \mathcal{B}_b}\sum_{1 \leq i \leq l, \boldsymbol{p}_i^n \text{ does not contain any } \boldsymbol{v}_s^*}^{l} G_{i,l+1}^n(\boldsymbol{w}^{(0)})
$$

$$
\cdot (\sum_{s=i+1}^{l+1}\sigma(\boldsymbol{w}^{(0)\top}\boldsymbol{p}_s^n)(\boldsymbol{\mu}_j^\top, 0^\top)\boldsymbol{p}_s^n) \tag{145}
$$

$$
\gtrsim - \frac{\eta(1-p_a)\beta^2}{M_1}.
$$

Hence, the conclusion holds when $t = 1$. Meanwhile, for any $k \in [M_2]$,

$$
(\boldsymbol{\nu}_k^\top, 0^\top)\boldsymbol{w}^{(1)} \leq \xi + \frac{\eta}{M_2}\sqrt{\frac{\log B}{B}}. \tag{146}
$$

Suppose that the conclusion holds when $t = t_0$ for $t_0 \lesssim \min\{\eta^{-1}\beta^{-2}\kappa_a^{-1}(1-p_a)^{-1}V, \eta^{-1}M_1^{\frac{2}{3}}\beta^{-\frac{2}{3}}\kappa_a^{-\frac{1}{3}}(1-p_a)^{-1}V^{\frac{1}{3}}\}$. Then, when $t = t_0 + 1$, we have that for $\boldsymbol{p}_s^n$ that does not contain any $\boldsymbol{v}_s^*, s \in [V]$

$$
-\frac{\eta(1-p_a)\beta^2 t_0}{M_1} - \sum_{i=1}^{t_0}i^2 \cdot (\frac{\eta^3(1-p_a)^3\beta^2}{M_1^3}) \lesssim \boldsymbol{w}^{(t_0)\top}\boldsymbol{p}_s^n \lesssim t_0 \cdot (-\frac{\eta\beta^2}{M_1} + \frac{\eta}{M_2}\sqrt{\frac{\log B}{B}} + \xi) < 0. \tag{147}
$$

For another $\boldsymbol{p}_r^n$, $r \neq s$, that contains a $\boldsymbol{v}_s^*$, $s \in [V]$,

$$\boldsymbol{w}^{(t_0)^\top} \boldsymbol{p}_r^n \lesssim t_0 \cdot (0 - \eta \beta^2 \kappa_a (1 - p_a)) < \boldsymbol{w}^{(t_0)^\top} \boldsymbol{p}_s^n < 0. \tag{148}$$

Then, with a high probability, we have for any $s \in [V]$,

$$
\begin{aligned}
&\boldsymbol{v}_s^{*\top} \boldsymbol{w}^{(t)} \\
=&\boldsymbol{v}_s^{*\top} (\boldsymbol{w}^{(t-1)} - \eta \frac{\partial \ell(\Psi; \boldsymbol{P}^n, z^n)}{\partial \boldsymbol{w}}) \\
\leq& - \eta \beta^2 t_0 \kappa_a (1 - p_a) - \eta \sum_{i=1}^{t_0-1} i^2 (\frac{\eta^2 (1 - p_a)^3 \beta^2}{M_1^2}) \kappa_a - \eta \frac{z^n}{|\mathcal{B}_1|} \sum_{n \in \mathcal{B}_b} \sum_{i=1}^{l} y_i^n (\beta^2 \\
&+ \frac{\eta^2 t_0^2 (1 - p_a)^2 \beta^2}{M_1^2}) G_{i,l+1}^n(\boldsymbol{w}^{(t_0)}) (\sum_{s=i+1}^{l+1} \sigma(\boldsymbol{w}^{(t_0)^\top} \boldsymbol{p}_s^n) \boldsymbol{v}_s^{*\top} \boldsymbol{p}_s^n - (1 - \sigma(\boldsymbol{w}^{(t_0)^\top} \boldsymbol{p}_i^n)) \boldsymbol{v}_s^{*\top} \boldsymbol{p}_i^n),
\end{aligned}
\tag{149}
$$

where the last step is by (110) and (127). Following our proof idea in the case of $t = 1$, we have that for $\boldsymbol{p}_i^n$ that contains a $\boldsymbol{v}_s^*$, $s \in [V]$, the corresponding $y_i^n$ has a probability of $1/2$ to be both binary labels. Then, by Hoeffding' bound (22), we have

$$
\begin{aligned}
&\Big\| \frac{\eta}{|\mathcal{B}_1|} \sum_{n \in \mathcal{B}_1} z^n \sum_{1 \leq i \leq l, \boldsymbol{p}_i^n \text{ contains } \boldsymbol{v}_s^*} y_i^n \boldsymbol{p}_i^{n\top} \boldsymbol{W}_B^{(t_0)^\top} \boldsymbol{W}_C^{(t_0)} \boldsymbol{p}_{query}^n G_{i,l+1}^n(\boldsymbol{w}^{(t_0)}) \\
&\cdot (\sum_{s=i+1}^{l+1} \sigma(\boldsymbol{w}^{(t_0)^\top} \boldsymbol{p}_s^n) \boldsymbol{p}_s^n - (1 - \sigma(\boldsymbol{w}^{(t_0)^\top} \boldsymbol{p}_i^n)) \boldsymbol{p}_i^n) \Big\| \\
\leq& \eta \sqrt{\frac{\log B}{B}}.
\end{aligned}
\tag{150}
$$

Then, with a high probability,

$$
\eta \frac{z^n}{|\mathcal{B}_1|} \sum_{n \in \mathcal{B}_b} \sum_{i=1}^{l} y_i^n (\beta^2 + \frac{\eta^2 t_0^2 (1-p_a)^2 \beta^2}{M_1^2}) G_{i,l+1}^n(\boldsymbol{w}^{(t_0)}) (\sum_{s=i+1}^{l+1} \sigma(\boldsymbol{w}^{(t_0)\top} \boldsymbol{p}_s^n) \boldsymbol{v}_s^{*\top} \boldsymbol{p}_s^n
$$

$$
\cdot -(1 - \sigma(\boldsymbol{w}^{(t_0)\top} \boldsymbol{p}_i^n)) \boldsymbol{v}_s^{*\top} \boldsymbol{p}_i^n)
$$

$$
\gtrsim -\eta \sqrt{\frac{\log B}{B}} + \eta \frac{z^n}{|\mathcal{B}_1|} \sum_{n \in \mathcal{B}_b} \sum_{\boldsymbol{p}_i^n \text{ does not contain } \boldsymbol{v}_s^*, z^n y_i^n = 1} y_i^n (\beta^2 + \frac{\eta^2 t_0^2 (1-p_a)^2 \beta^2}{M_1^2})
$$

$$
\cdot G_{i,l+1}^n(\boldsymbol{w}^{(t_0)}) (\sum_{s=i+1}^{l+1} \sigma(\boldsymbol{w}^{(t_0)\top} \boldsymbol{p}_s^n) \boldsymbol{v}_s^{*\top} \boldsymbol{p}_s^n - (1 - \sigma(\boldsymbol{w}^{(t_0)\top} \boldsymbol{p}_i^n)) \boldsymbol{v}_s^{*\top} \boldsymbol{p}_i^n)
$$

$$
= -\eta \sqrt{\frac{\log B}{B}} + \eta \frac{1}{|\mathcal{B}_1|} \sum_{n \in \mathcal{B}_b} \sum_{\boldsymbol{p}_i^n \text{ does not contain } \boldsymbol{v}_s^*} (\beta^2 + \frac{\eta^2 t_0^2 (1-p_a)^2 \beta^2}{M_1^2}) G_{i,l+1}^n(\boldsymbol{w}^{(t_0)})
$$

$$
\cdot \sum_{s=i+1}^{l+1} \sigma(\boldsymbol{w}^{(t_0)\top} \boldsymbol{p}_s^n) \boldsymbol{v}_s^{*\top} \boldsymbol{p}_s^n \tag{151}
$$

$$
\gtrsim -\eta \sqrt{\frac{\log B}{B}} + \eta \frac{1}{|\mathcal{B}_1|} \sum_{n \in \mathcal{B}_b} \sum_{\boldsymbol{p}_i^n \text{ does not contain } \boldsymbol{v}_s^*} (\beta^2 + \frac{\eta^2 t_0^2 (1-p_a)^2 \beta^2}{M_1^2}) G_{i,l+1}^n(\boldsymbol{w}^{(t_0)})
$$

$$
\cdot (l-i+1) \frac{\kappa_a}{V}
$$

$$
\gtrsim -\eta \sqrt{\frac{\log B}{B}} + \eta (\beta^2 + \frac{\eta^2 t_0^2 (1-p_a)^2 \beta^2}{M_1^2}) \mathbb{E} \left[ \sum_{i=1}^{l} G_{i,l+1}^n(\boldsymbol{w}^{(t_0)}) (l-i+1) \frac{\kappa_a (1-p_a)}{V} \right]
$$

$$
\gtrsim -\eta \sqrt{\frac{\log B}{B}} + \eta (\beta^2 + \frac{\eta^2 t_0^2 (1-p_a)^2 \beta^2}{M_1^2}) \mathbb{E} \left[ \sum_{i=1}^{l} G_{i,l+1}^n(\boldsymbol{w}^{(t_0)}) \frac{\kappa_a (1-p_a)}{V} \right]
$$

$$
\geq -\eta \sqrt{\frac{\log B}{B}} + \eta (\beta^2 + \frac{\eta^2 t_0^2 (1-p_a)^2 \beta^2}{M_1^2}) \frac{\kappa_a (1-p_a)}{V},
$$

where the fourth step follows the idea of (139) since

$$
G_{i,l+1}^n(\boldsymbol{w}^{(t_0)}) (l-i+1) \leq \Theta(1), \tag{152}
$$

for any $i \in [l]$ and $n \in \mathcal{B}_b$. The last step of (151) follows from

$$
\sum_{i=1}^{l} G_{i,l+1}^n(\boldsymbol{w}^{(t_0)}) = 1 - \sigma(\boldsymbol{w}^{(t_0)\top} \boldsymbol{p}_{query}^n) - \prod_{i=1}^{l+1} (1 - \sigma(\boldsymbol{w}^{(t_0)\top} \boldsymbol{p}_i^n)) \geq \frac{1}{4}, \tag{153}
$$

since

$$
\sigma(\boldsymbol{w}^{(t_0)\top} \boldsymbol{p}_{query}^n) < \sigma(0) = \frac{1}{2}, \tag{154}
$$

by (147), and with a high probability,

$$
\prod_{i=1}^{l+1} (1 - \sigma(\boldsymbol{w}^{(t_0)\top} \boldsymbol{p}_i^n)) \leq \prod_{\boldsymbol{p}_i^n \text{ does not contain any } \boldsymbol{v}_s^*} (1 - \sigma(\boldsymbol{w}^{(t_0)\top} \boldsymbol{p}_i^n))
$$

$$
\lesssim (1 - \frac{1}{1 + e^{-\frac{V}{\kappa_a M_1}}})^{l(1-p_a)} \tag{155}
$$

$$
\leq \frac{1}{4},
$$

where the last step holds if

$$
l \gtrsim (1-p_a)^{-1} \log M_1. \tag{156}
$$

The second step of (155) comes from (147) and

$$\Pr\left(\left|\frac{1}{l}\sum_{i=1}^{l}\mathbb{1}[\boldsymbol{p}_i^n \text{ does not contain } \boldsymbol{v}_s^*] - (1-p_a)\right| \geq c \cdot (1-p_a)\right) \lesssim e^{-l(1-p_a)c^2} \leq M_1^{-C} \tag{157}$$

by Lemma 1 for some $c \in (0,1), C > 1$, and

$$l \geq (1-p_a)^{-1}\log M_1. \tag{158}$$

Then, by plugging (151) into (149), we have

$$\begin{aligned}
&\boldsymbol{v}_s^{*\top}\boldsymbol{w}^{(t_0+1)}\\
&\leq -\frac{\eta\beta^2 t_0\kappa_a(1-p_a)}{V} - \eta\sum_{i=1}^{t_0-1}i^2\left(\frac{\eta^2(1-p_a)^3\beta^2}{M_1^2}\right)\frac{\kappa_a}{V} + \eta\sqrt{\frac{\log B}{B}} - \eta(\beta^2\\
&\quad + \frac{\eta^2 t_0^2(1-p_a)^2\beta^2}{M_1^2}) \cdot \frac{\kappa_a(1-p_a)}{V}\\
&= -\frac{\eta\beta^2(t_0+1)\kappa_a(1-p_a)}{V} - \eta\sum_{i=1}^{t_0}i^2\left(\frac{\eta^2(1-p_a)^3\beta^2}{M_1^2}\right)\frac{\kappa_a}{V} + \eta\sqrt{\frac{\log B}{B}}\\
&\lesssim -\frac{\eta\beta^2(t_0+1)\kappa_a(1-p_a)}{V} - \eta\sum_{i=1}^{t_0}i^2\left(\frac{\eta^2(1-p_a)^3\beta^2}{M_1^2}\right)\frac{\kappa_a}{V},
\end{aligned} \tag{159}$$

where the last step holds given (141) and $t_0 \lesssim \min\{\eta^{-1}\beta^{-2}\kappa_a^{-1}(1-p_a)^{-1}V, \eta^{-1}M_1^{\frac{2}{3}}\beta^{-\frac{2}{3}}\kappa_a^{-\frac{1}{3}}(1-p_a)^{-1}V^{\frac{1}{3}}\}$. We can also derive that for any $j \in [M_1]$,

$$\begin{aligned}
&(\boldsymbol{\mu}_j^\top, 0^\top)\boldsymbol{w}^{(t)}\\
&\leq \xi + \frac{\eta}{M_1}\sqrt{\frac{\log B}{B}} - \frac{\eta(1-p_a)\beta^2 t_0}{M_1} - \sum_{i=1}^{t_0-1}i^2 \cdot \left(\frac{\eta^3(1-p_a)^3\beta^2}{M_1^3}\right) - \frac{\eta}{|\mathcal{B}_1|}\sum_{n\in\mathcal{B}_b}\\
&\quad \sum_{\boldsymbol{p}_i^n \text{ does not contain any } \boldsymbol{v}_s^*}^{l}(\beta^2 + \frac{\eta^2 t_0^2(1-p_a)^2\beta^2}{M_1^2})G_{i,l+1}^n(\boldsymbol{w}^{(t_0)}) \cdot \left(\sum_{s=i+1}^{l+1}\sigma(\boldsymbol{w}^{(t_0)\top}\boldsymbol{p}_s^n)\right)\\
&\quad \cdot (\boldsymbol{\mu}_j^\top, 0^\top)\boldsymbol{p}_s^n - (1-\sigma(\boldsymbol{w}^{(t_0)\top}\boldsymbol{p}_i^n))(\boldsymbol{\mu}_j^\top, 0^\top)\boldsymbol{p}_i^n\\
&\lesssim \xi + \frac{\eta}{M_1}\sqrt{\frac{\log B}{B}} - \frac{\eta(1-p_a)\beta^2 t_0}{M_1} - \sum_{i=1}^{t_0-1}i^2 \cdot \left(\frac{\eta^3(1-p_a)^3\beta^2}{M_1^3}\right) - \frac{\eta}{|\mathcal{B}_1|}\sum_{n\in\mathcal{B}_b}\sum_{i=1}^{l}(\beta^2\\
&\quad + \frac{\eta^2 t_0^2(1-p_a)^2\beta^2}{M_1^2}) \cdot G_{i,l+1}^n(\boldsymbol{w}^{(t_0)})(l-i+1) \cdot \frac{(1-p_a)}{M_1}\\
&\lesssim \xi + \frac{\eta}{M_1}\sqrt{\frac{\log B}{B}} - \frac{\eta(1-p_a)\beta^2 t_0}{M_1} - \sum_{i=1}^{t_0-1}i^2 \cdot \left(\frac{\eta^3(1-p_a)^3\beta^2}{M_1^3}\right) - \eta\frac{(1-p_a)}{M_1}(\beta^2\\
&\quad + \frac{\eta^2 t_0^2(1-p_a)^2\beta^2}{M_1^2})\\
&\lesssim \xi + \frac{\eta}{M_1}\sqrt{\frac{\log B}{B}} - \frac{\eta(1-p_a)\beta^2(t_0+1)}{M_1} - \sum_{i=1}^{t_0-1}i^2 \cdot \left(\frac{\eta^3(1-p_a)^3\beta^2}{M_1^3}\right)\\
&\quad - \frac{\eta(1-p_a)}{M_1}\left(\frac{\eta^2 t_0^2(1-p_a)^2\beta^2}{M_1^2}\right)\\
&\lesssim -\frac{\eta(1-p_a)\beta^2(t_0+1)}{M_1} - \sum_{i=1}^{t_0}i^2 \cdot \left(\frac{\eta^3(1-p_a)^3\beta^2}{M_1^3}\right),
\end{aligned} \tag{160}$$

where the second step of (160) follows the second step in (142) using Lemma 2. Meanwhile,

$$
\begin{aligned}
(\boldsymbol{\mu}_j^\top, 0^\top)\boldsymbol{w}^{(t)} & \\
\gtrsim & -\xi - \frac{\eta}{M_1}\sqrt{\frac{\log B}{B}} - \frac{\eta(1-p_a)\beta^2 t_0}{M_1} - \sum_{i=1}^{t_0-1} i^2 \cdot \left(\frac{\eta^3(1-p_a)^3\beta^2}{M_1^3}\right) - \frac{\eta}{|\mathcal{B}_1|}\sum_{n\in\mathcal{B}_b}\sum_{i=1}^{l}(\beta^2 \\
& + \frac{\eta^2 t_0^2(1-p_a)^2\beta^2}{M_1^2}) \cdot G_{i,l+1}^n(\boldsymbol{w}^{(t_0)})(l-i+1)\cdot\frac{(1-p_a)}{M_1} \\
\gtrsim & -\xi - \frac{\eta}{M_1}\sqrt{\frac{\log B}{B}} - \frac{\eta(1-p_a)\beta^2 t_0}{M_1} - \sum_{i=1}^{t_0-1} i^2\cdot\left(\frac{\eta^3(1-p_a)^3\beta^2}{M_1^3}\right) - \eta\frac{(1-p_a)}{M_1}(\beta^2 \\
& + \frac{\eta^2 t_0^2(1-p_a)^2\beta^2}{M_1^2}) \\
\gtrsim & -\frac{\eta(1-p_a)\beta^2(t_0+1)}{M_1} - \sum_{i=1}^{t_0} i^2\cdot\left(\frac{\eta^3(1-p_a)^3\beta^2}{M_1^3}\right),
\end{aligned}
\tag{161}
$$

where the second step is by Lemma 6. Therefore, we complete the induction.

□

### F.4. Proof of Lemma 5

*Proof.* Let

$$
t_0 = \Theta(\eta^{-1}(1-p_a)^{-1}\beta^{-2}M_1).
\tag{162}
$$

(a) We first prove that for any $s \in [V]$,

$$
(\boldsymbol{v}_s^{*\top}, 0^\top)\boldsymbol{w}^{(t)} \leq \Theta(-\log(2+t\gamma_1))
\tag{163}
$$

for some $\gamma_1 > 0$ by induction. When $t = \min\{\eta^{-1}\beta^{-2}\kappa_a^{-1}(1-p_a)^{-1}V, \eta^{-1}M_1^{\frac{2}{3}}\beta^{-\frac{2}{3}}\kappa_a^{-\frac{1}{3}}(1-p_a)^{-1}V^{\frac{1}{3}}\}$, we have

$$
(\boldsymbol{v}_s^{*\top}, 0^\top)\boldsymbol{w}^{(t)} \lesssim -\Theta(1) \leq \Theta(-\log(2+\eta^{-1}\beta^{-\frac{2}{3}}\kappa_a^{-\frac{1}{3}}M_1^{\frac{2}{3}}(1-p_a)^{-1}V^{\frac{1}{3}}\gamma_1))
\tag{164}
$$

by Lemma 4 for any $\gamma_1 > 0$, since that $1 + \eta^{-1}\beta^{-\frac{2}{3}}\kappa_a^{-\frac{1}{3}}M_1^{\frac{2}{3}}(1-p_a)^{-1}V^{\frac{1}{3}}\gamma_1 \geq \Theta(1)$ and $\gamma_1 > 0$. Therefore, (163) holds when

$$
t = \min\{\eta^{-1}\beta^{-2}\kappa_a^{-1}(1-p_a)^{-1}V, \eta^{-1}M_1^{\frac{2}{3}}\beta^{-\frac{2}{3}}\kappa_a^{-\frac{1}{3}}(1-p_a)^{-1}V^{\frac{1}{3}}\}.
\tag{165}
$$

Suppose that when $t \leq t_2$ with $t_2 > \min\{\eta^{-1}\beta^{-2}\kappa_a^{-1}(1-p_a)^{-1}V, \eta^{-1}M_1^{\frac{2}{3}}\beta^{-\frac{2}{3}}\kappa_a^{-\frac{1}{3}}(1-p_a)^{-1}V^{\frac{1}{3}}\}$ and $t_2 \leq t_0$, the conclusion still holds. Then, when $t = t_2 + 1$, we have

$$
\begin{aligned}
(\boldsymbol{v}_s^{*\top}, 0^\top)\boldsymbol{w}^{(t)} \lesssim & -\log(2+t_2\gamma_1) - \frac{\eta(1-p_a)\kappa_a}{V}(\beta^2 + \frac{\eta^2 t_2^2(1-p_a)^2\beta^2}{M_1^2})\cdot\frac{1}{1+e^{\log(2+t_2\gamma_1)}} \\
= & -\log(2+t_2\gamma_1) - \frac{\eta(1-p_a)\kappa_a}{V}(\beta^2 + \frac{\eta^2 t_2^2(1-p_a)^2\beta^2}{M_1^2})\cdot(3+t_2\gamma_1)^{-1} \\
\lesssim & -\log(2+(t_2+1)\gamma_1),
\end{aligned}
\tag{166}
$$

where the last step comes from the following.
(i)

$$
\begin{aligned}
\frac{\eta(1-p_a)\beta^2\kappa_a}{V}(3+t_2\gamma_1)^{-1} & \gtrsim \log(1 + \frac{\gamma_1}{2+t_2\gamma_1}) \\
& = \log(2+(t_2+1)\gamma_1) - \log(2+t_2\gamma_1),
\end{aligned}
\tag{167}
$$

where the first step is from

$$
\gamma_1 \leq \eta(1-p_a)\beta^2.
\tag{168}
$$

(ii)

$$\eta^3 \frac{(1-p_a)^3 \kappa_a}{M_1^2 V} \beta^2 t_2^2 (3 + t_2 \gamma_1)^{-1} \gtrsim \log(2 + (t_2 + 1)\gamma_1) - \log(2 + t_2 \gamma_1), \tag{169}$$

which comes from

$$\gamma_1 \leq \frac{\eta(1-p_a)\beta^{-2}\kappa_a}{V}. \tag{170}$$

Therefore, (163) can be rewritten as

$$({\boldsymbol{v}_s^*}^\top, 0^\top)\boldsymbol{w}^{(t)} \leq \Theta(-\log(2 + t \cdot \eta(1-p_a)\beta^2)), \tag{171}$$

when $\kappa_a \geq V\beta^{-4}$, so that the conclusion holds when $t = t_2 + 1$. Thus, the induction can be completed. We can then derive that when $t = t_0$, we have

$$({\boldsymbol{v}_s^*}^\top, 0^\top)\boldsymbol{w}^{(t_0)} \leq \Theta(-\log(2 + t_0 \cdot \eta(1-p_a)\beta^2)) \lesssim -\log(M_1), \tag{172}$$

and for $\boldsymbol{p}_i$ that contains $\boldsymbol{\nu}_*$,

$$\sigma(\boldsymbol{p}_i^\top \boldsymbol{w}^{(t)}) \lesssim \frac{1}{\mathrm{poly}(M_1)}. \tag{173}$$

(b) We then prove that

$$(\boldsymbol{\mu}_j^\top, 0^\top)\boldsymbol{w}^{(t)} \geq \Theta(-\log(2 + \frac{t\gamma_2}{M_1})) \tag{174}$$

for $j \in [M_1]$ and some $\gamma_2 > 0$ by induction. When $t = \min\{\eta^{-1}\beta^{-2}\kappa_a^{-1}(1-p_a)^{-1}V, \eta^{-1}M_1^{\frac{2}{3}}\beta^{-\frac{2}{3}}\kappa_a^{-\frac{1}{3}}(1-p_a)^{-1}V^{\frac{1}{3}}\}$, we have

$$(\boldsymbol{\mu}_j^\top, 0^\top)\boldsymbol{w}^{(t)} \gtrsim -\frac{1}{M_1} \geq \Theta(-\log(2 + \eta^{-1}\beta^{-\frac{2}{3}}\kappa_a^{-\frac{1}{3}}M_1^{-\frac{1}{3}}(1-p_a)^{-1}V^{\frac{1}{3}}\gamma_2)) \tag{175}$$

by Lemma 4 for any $\gamma_2 > 0$, since that $1 + \eta^{-1}\beta^{-\frac{2}{3}}\kappa_a^{-\frac{1}{3}}M_1^{-\frac{1}{3}}(1-p_a)^{-1}V^{\frac{1}{3}}\gamma_2 \gg M_1^{-1}$ and $\gamma_2 \geq 1$. Therefore, (174) holds when

$$t = \min\{\eta^{-1}\beta^{-2}\kappa_a^{-1}(1-p_a)^{-1}V, \eta^{-1}M_1^{\frac{2}{3}}\beta^{-\frac{2}{3}}\kappa_a^{-\frac{1}{3}}(1-p_a)^{-1}V^{\frac{1}{3}}\}. \tag{176}$$

Suppose that when $t \leq t_2$ with $t_2 > \min\{\eta^{-1}\beta^{-2}\kappa_a^{-1}(1-p_a)^{-1}V, \eta^{-1}M_1^{\frac{2}{3}}\beta^{-\frac{2}{3}}\kappa_a^{-\frac{1}{3}}(1-p_a)^{-1}V^{\frac{1}{3}}\}$ and $t_2 \leq t_0$, the conclusion still holds. Then, when $t = t_2 + 1$, we have

$$
\begin{aligned}
&(\boldsymbol{\mu}_j^\top, 0^\top)\boldsymbol{w}^{(t)} \\
&\gtrsim -\log(2 + \frac{t_2\gamma_2}{M_1}) - \eta\frac{(1-p_a)}{M_1}(\beta^2 + \frac{\eta^2 t_2^2(1-p_a)^2\beta^2}{M_1^2}) \cdot \frac{1}{1 + e^{\log(2 + \frac{t_2\gamma_2}{M_1})}} \\
&= -\log(2 + \frac{t_2\gamma_2}{M_1}) - \eta\frac{(1-p_a)}{M_1}(\beta^2 + \frac{\eta^2 t_2^2(1-p_a)^2\beta^2}{M_1^2}) \cdot (3 + \frac{t_2\gamma_2}{M_1})^{-1} \\
&\gtrsim -\log(2 + \frac{(t_2+1)\gamma_2}{M_1}),
\end{aligned}
\tag{177}
$$

where the last step comes from the following.
(i)

$$
\begin{aligned}
\eta\frac{(1-p_a)}{M_1}\beta^2(3 + \frac{t_2\gamma_2}{M_1})^{-1} &\lesssim \log(1 + \frac{\frac{\gamma_2}{M_1}}{2 + \frac{t_2\gamma_2}{M_1}}) \\
&= \log(2 + \frac{(t_2+1)\gamma_2}{M_1}) - \log(2 + \frac{t_2\gamma_2}{M_1}),
\end{aligned}
\tag{178}
$$

where the first step is from

$$\gamma_2 \geq \eta(1-p_a)\beta^2. \tag{179}$$

(ii)

$$\eta^3 \frac{(1-p_a)^3}{M_1^3}\beta^2 t_2^2(3 + \frac{t_2\gamma_2}{M_1})^{-1} \lesssim \log(2 + \frac{(t_2+1)\gamma_2}{M_1}) - \log(2 + \frac{t_2\gamma_2}{M_1}), \tag{180}$$

which comes from

$$\gamma_2 \geq \eta(1 - p_a)\beta^{-2}. \tag{181}$$

Therefore, (174) can be rewritten as

$$(\boldsymbol{\mu}_j^\top, \boldsymbol{0}^\top)\boldsymbol{w}^{(t)} \geq \Theta(-\log(2 + t \cdot \frac{\eta(1 - p_a)\beta^2}{M_1})), \tag{182}$$

so that the conclusion holds when $t = t_2 + 1$. Thus, the induction can be completed. We can then derive that when $t = t_0$, we have

$$(\boldsymbol{\mu}_j^\top, \boldsymbol{0}^\top)\boldsymbol{w}^{(t_0)} \geq \Theta(-\log(2 + t_0 \cdot \frac{\eta(1 - p_a)\beta^2}{M_1})) \geq -\log(3) \geq -\Theta(1), \tag{183}$$

and for $\boldsymbol{p}_i$ that does not contain $\boldsymbol{\nu}_*$,

$$\sigma(\boldsymbol{p}_i^\top \boldsymbol{w}^{(t)}) \gtrsim \Theta(1). \tag{184}$$

$\square$

## F.5. Proof of Lemma 6

*Proof.* Given a prompt $\boldsymbol{P}$ defined in (2) with $(\boldsymbol{x}_1, \boldsymbol{x}_2, \cdots, \boldsymbol{x}_l, \boldsymbol{x}_{query})$, let $\boldsymbol{x}_{l+1} = \boldsymbol{x}_{query}$. Define

$$
\begin{aligned}
\hat{\boldsymbol{P}}^i &= \begin{pmatrix} \boldsymbol{x}_{i+1} & \boldsymbol{x}_{i+2} & \cdots & \boldsymbol{x}_l & \boldsymbol{x}_{l+1} & \boldsymbol{x}_1 & \boldsymbol{x}_2 & \cdots & \boldsymbol{x}_i \\ y_{i+1} & y_{i+2} & \cdots & y_l & y_{l+1} & y_1 & y_2 & \cdots & y_i \end{pmatrix} \\
&:= \begin{pmatrix} \hat{\boldsymbol{x}}_1^i & \hat{\boldsymbol{x}}_2^i & \cdots & \hat{\boldsymbol{x}}_l^i & \hat{\boldsymbol{x}}_{l+1}^i \\ \hat{y}_1^i & \hat{y}_2^i & \cdots & \hat{y}_l^i & \hat{y}_{l+1}^i \end{pmatrix} \\
&:= (\hat{\boldsymbol{p}}_1^i, \hat{\boldsymbol{p}}_2^i, \cdots, \hat{\boldsymbol{p}}_l^i, \hat{\boldsymbol{p}}_{l+1}^i),
\end{aligned} \tag{185}
$$

which is a rotation of in-context examples for $i \in [l] \cup \{0\}$. Therefore, we have

$$
\begin{aligned}
&\sum_{i=1}^l G_{i,l+1}(\boldsymbol{w}^{(t)})(l - i + 1) \\
=& \sum_{i=1}^l G_{i,l+1}^0(\boldsymbol{w}^{(t)})(l - i + 1) \\
\leq& \sum_{i=1}^l G_{i,l+1}^0(\boldsymbol{w}^{(t)}) + \sum_{i=1}^l G_{i,l+1}^l(\boldsymbol{w}^{(t)})(1 - \sigma(\boldsymbol{w}^{(t)\top}\hat{\boldsymbol{p}}_1^l)) + \sum_{i=1}^l G_{i,l+1}^{l-1}(\boldsymbol{w}^{(t)})(1 \\
& - \sigma(\boldsymbol{w}^{(t)\top}\hat{\boldsymbol{p}}_1^{l-1}))(1 - \sigma(\boldsymbol{w}^{(t)\top}\hat{\boldsymbol{p}}_2^{l-1})) + \cdots + \sum_{i=1}^l G_{i,l+1}^2(\boldsymbol{w}^{(t)}) \prod_{j=1}^{l-1}(1 - \sigma(\boldsymbol{w}^{(t)\top}\hat{\boldsymbol{p}}_j^2)) \\
\leq& \max_{j \in [l]} \left\{ \sum_{i=1}^l G_{i,l+1}^j(\boldsymbol{w}^{(t)}) \right\} \cdot (1 + (1 - \sigma(\boldsymbol{w}^{(t)\top}\hat{\boldsymbol{p}}_1^l)) + (1 - \sigma(\boldsymbol{w}^{(t)\top}\hat{\boldsymbol{p}}_1^{l-1}))(1 \\
& - \sigma(\boldsymbol{w}^{(t)\top}\hat{\boldsymbol{p}}_2^{l-1})) + \cdots + \prod_{j=1}^{l-1}(1 - \sigma(\boldsymbol{w}^{(t)\top}\hat{\boldsymbol{p}}_j^2))) \\
\leq& 1 + (1 - \sigma(\boldsymbol{w}^{(t)\top}\hat{\boldsymbol{p}}_1^l)) + (1 - \sigma(\boldsymbol{w}^{(t)\top}\hat{\boldsymbol{p}}_1^{l-1}))(1 - \sigma(\boldsymbol{w}^{(t)\top}\hat{\boldsymbol{p}}_2^{l-1})) + \cdots \\
& + \prod_{j=1}^{l-1}(1 - \sigma(\boldsymbol{w}^{(t)\top}\hat{\boldsymbol{p}}_j^2)) \\
\leq& 1 + 1 - c + (1 - c)^2 + \cdots + (1 - c)^{l-1} \\
\leq& \frac{1}{c} \\
\leq& \Theta(1),
\end{aligned} \tag{186}
$$

where the third to last step holds since that when $t \lesssim \min\{\eta^{-1}\beta^{-2}\kappa_a^{-1}(1-p_a)^{-1}V, \eta^{-1}M_1^{\frac{2}{3}}((1-p_a)\beta)^{-\frac{2}{3}}(\kappa_a(1-p_a))^{-\frac{1}{3}}V^{\frac{1}{3}}\}$, there exists $c \in (0,1)$ and $C \in (0,1)$, $C > c$, such that $c \leq \sigma(\boldsymbol{w}^{(t)^{\top}}\boldsymbol{p}_j) \leq C$ for any $j \in [l]$. $\qquad\square$

## G. Extension to Other SSM/Linear RNN Architectures

Our theoretical analysis can be extended to a broader range of SSM or Linear RNN architectures. The key to such extension depends on whether the basic block of the model can be decomposed into a linear attention layer and a gating layer as in (3). Even if the specific form of the nonlinear gating differs from that in the Mamba architecture we consider in this work, we can still compute the gradient of the new gating function and analyze the resulting training dynamics and generalization performance. We then list several examples and briefly discuss how their models can be interpreted as linear attention plus a gating based on the summary from Table 2 of (Yang et al., 2024c).

- **Mamba-2 (Dao & Gu, 2024)**. The updating equation of Mamba-2 is

$$
\begin{aligned}
\boldsymbol{h}_i &= \gamma(\boldsymbol{w}, a; i) \cdot \boldsymbol{h}_{i-1} + \boldsymbol{v}_i \boldsymbol{k}_i^{\top} \quad \in \mathbb{R}^{d_0 \times m}, \quad \forall i \in [m] \\
\boldsymbol{o}_i &= \boldsymbol{h}_i \boldsymbol{q}_i \quad \in \mathbb{R}^{d_0},
\end{aligned}
\tag{187}
$$

where $\gamma(\boldsymbol{w}, a; i) = e^{-\text{softplus}(\boldsymbol{w}^{\top}\boldsymbol{p}_i)e^a} \in \mathbb{R}$ for $a \in \mathbb{R}$ and $\boldsymbol{w} \in \mathbb{R}^{d_0}$ from Table 1 of (Yang et al., 2024b). Then,

$$
\begin{aligned}
\boldsymbol{h}_t &= \gamma(\boldsymbol{w}, a; t) \cdot \boldsymbol{h}_{t-1} + \boldsymbol{v}_t \boldsymbol{k}_t^{\top} \\
&= \gamma(\boldsymbol{w}, a; t) \cdot (\gamma(\boldsymbol{w}, a; t-1) \cdot \boldsymbol{h}_{t-2} + \boldsymbol{v}_{t-1} \boldsymbol{k}_{t-1}^{\top}) + \boldsymbol{v}_t \boldsymbol{k}_i^{\top} \\
&= \cdots \\
&:= \sum_{i=1}^{t} G_{i,t}(\boldsymbol{w}, a) \boldsymbol{v}_i \boldsymbol{k}_i^{\top},
\end{aligned}
\tag{188}
$$

where

$$
G_{i,t}(\boldsymbol{w}, a) = \begin{cases} \prod_{j=i+1}^{t} \gamma(\boldsymbol{w}, a; j), & i < t \\ 1, & i = t. \end{cases}
\tag{189}
$$

Therefore, the output of a Mamba-2 block can be written as a summation of linear attention output $\boldsymbol{v}_t \boldsymbol{k}_i^{\top} \boldsymbol{q}_i$ weighted by the scalar gating $G_{i,t}(\boldsymbol{w}, a)$ for $1 \leq i \leq t$.

- **RetNet (Sun et al., 2023)**. The updating equation of RetNet is

$$
\begin{aligned}
\boldsymbol{h}_i &= \gamma \cdot \boldsymbol{h}_{i-1} + \boldsymbol{v}_i \boldsymbol{k}_i^{\top} \quad \in \mathbb{R}^{d_0 \times m}, \quad \forall i \in [m] \\
\boldsymbol{o}_i &= \boldsymbol{h}_i \boldsymbol{q}_i \quad \in \mathbb{R}^{d_0}.
\end{aligned}
\tag{190}
$$

Then,

$$
\boldsymbol{h}_t = \gamma \cdot \boldsymbol{h}_{t-1} + \boldsymbol{v}_t \boldsymbol{k}_t^{\top} := \sum_{i=1}^{t} G_{i,t}(\boldsymbol{W}) \boldsymbol{v}_i \boldsymbol{k}_i^{\top},
\tag{191}
$$

where

$$
G_{i,t}(\boldsymbol{W}) = \begin{cases} \gamma^{t-i}, & i < t \\ 1, & i = t. \end{cases}
\tag{192}
$$

- **Gated Retention (Sun et al., 2024)**. The updating equation of Gated Retention is

$$
\begin{aligned}
\boldsymbol{h}_i &= \gamma(\boldsymbol{w}; i) \cdot \boldsymbol{h}_{i-1} + \boldsymbol{v}_i \boldsymbol{k}_i^{\top} \quad \in \mathbb{R}^{d_0 \times m}, \quad \forall i \in [m] \\
\boldsymbol{o}_i &= \boldsymbol{h}_i \boldsymbol{q}_i \quad \in \mathbb{R}^{d_0},
\end{aligned}
\tag{193}
$$

where $\gamma(\boldsymbol{w}; i) = \sigma(\boldsymbol{w}^{\top}\boldsymbol{p}_i)^{\frac{1}{\tau}} \in \mathbb{R}$ for $\tau \in \mathbb{R}$. Then,

$$
\boldsymbol{h}_t = \gamma(\boldsymbol{w}; i) \cdot \boldsymbol{h}_{t-1} + \boldsymbol{v}_t \boldsymbol{k}_t^{\top} := \sum_{i=1}^{t} G_{i,t}(\boldsymbol{W}) \boldsymbol{v}_i \boldsymbol{k}_i^{\top},
\tag{194}
$$

where

$$G_{i,t}(\boldsymbol{W}) = \begin{cases} \prod_{j=i+1}^{t} \gamma(\boldsymbol{w};j), & i < t \\ 1, & i = t. \end{cases} \tag{195}$$

- **Gated Linear Attention** (Yang et al., 2024b). The updating equation of Gated Linear Attention is

$$\begin{aligned} \boldsymbol{h}_i &= \boldsymbol{h}_{i-1} \odot (\sigma(\boldsymbol{W}\boldsymbol{p}_i)^{\frac{1}{\tau}} \mathbf{1}_m^{\top}) + \boldsymbol{v}_i \boldsymbol{k}_i^{\top} \quad \in \mathbb{R}^{d_0 \times m}, \quad \forall i \in [m] \\ \boldsymbol{o}_i &= \boldsymbol{h}_i \boldsymbol{q}_i \quad \in \mathbb{R}^{d_0}, \end{aligned} \tag{196}$$

where $\boldsymbol{W} \in \mathbb{R}^{d_0 \times d_0}$ for $\tau \in \mathbb{R}$. Then,

$$\boldsymbol{h}_t := \sum_{i=1}^{t} \boldsymbol{v}_i (\boldsymbol{k}_i \odot \sigma(\boldsymbol{W}\boldsymbol{u}_i)^{\frac{1}{\tau}})^{\top}, \tag{197}$$

$$F(\Psi; \boldsymbol{P}) = \sum_{i=1}^{t} y_i (\boldsymbol{W}_K \boldsymbol{p}_i \odot \sigma(\boldsymbol{W}\boldsymbol{p}_i)^{\frac{1}{\tau}})^{\top} \boldsymbol{W}_Q \boldsymbol{p}_{query}. \tag{198}$$

Note that in this case, the gating is essentially applied to the key rather than the value as in our (3). Then,

$$\frac{\partial F(\Psi; \boldsymbol{P})}{\partial \boldsymbol{W}} = \sum_{i=1}^{t} y_i (\boldsymbol{W}_K \boldsymbol{p}_i \odot \boldsymbol{W}_Q \boldsymbol{p}_{query}) \odot \frac{1}{\tau} \sigma(\boldsymbol{W}\boldsymbol{p}_i)^{\frac{1}{\tau}} \odot (1 - \sigma(\boldsymbol{W}\boldsymbol{p}_i)) \boldsymbol{p}_i^{\top}. \tag{199}$$

Our gradient analysis is to characterize the feature updates of (199).

## H. Extension to Multi-Classification Problems

Our theoretical analysis can be extended from binary classification to a basic setting of multi-classification problems. For a $C$-classification problem, where $C = 2^H$ for a certain integer $S > 0$, we can decompose this classification problem into an $H$-level hierarchical classification task, where each level is a binary classification problem. Correspondingly, we assume that the labels of the context examples and the query are $H$-dimensional, i.e., $\boldsymbol{z}, \boldsymbol{y}_h \in \{+1, -1\}^H$, $h \in [H]$. We assume that each context input $\boldsymbol{x} \in \mathbb{R}^{d_H}$, where $d_H = d \cdot H$. Denote $\boldsymbol{x}_h$ as the coordinates from $d_0(h-1) + 1$ to $d_0 h$ of $\boldsymbol{x}$. The formulation of $\boldsymbol{x}_h$ follows the definition in (6). Then, the prompt $\boldsymbol{P} \in \mathbb{R}^{((d+1)H) \times (l+1)}$ for $\boldsymbol{x}_{query}$ is constructed as

$$\boldsymbol{P} = (\boldsymbol{P}_1^{\top}, \boldsymbol{P}_2^{\top}, \cdots, \boldsymbol{P}_H^{\top})^{\top}, \; \boldsymbol{P}_h = \begin{pmatrix} \boldsymbol{x}_{1,h} & \boldsymbol{x}_{2,h} & \cdots & \boldsymbol{x}_{l,h} & \boldsymbol{x}_{query,h} \\ \boldsymbol{y}_{1,h} & \boldsymbol{y}_{2,h} & \cdots & \boldsymbol{y}_{l,h} & 0 \end{pmatrix} \in \mathbb{R}^{(d+1) \times (l+1)}. \tag{200}$$

Then, we can consider an $H$-head Mamba model parameterized by $\Psi = \{\{\boldsymbol{W}_{B,h}, \boldsymbol{W}_{C,h}, \boldsymbol{w}_h\}_{h=1}^{H}\}$. Following (3), the output of one-layer Mamba can be rewritten as

$$\begin{aligned} F(\Psi; \boldsymbol{P}) &= (F_1(\Psi; \boldsymbol{P}), F_2(\Psi; \boldsymbol{P}) \cdots, F_H(\Psi; \boldsymbol{P}))^{\top}, \\ F_h(\Psi; \boldsymbol{P}) &= \sum_{i=1}^{l+1} G_{i,l+1}(\boldsymbol{w}_h) y_{i,h} \boldsymbol{p}_{i.h}^{\top} \boldsymbol{W}_{B,h}^{\top} \boldsymbol{W}_{C,h} \boldsymbol{p}_{query,h}, \\ \text{where } G_{i,l+1}(\boldsymbol{w}_h) &= \begin{cases} \sigma(\boldsymbol{w}_h^{\top} \boldsymbol{p}_{i,h}) \prod_{j=i+1}^{l+1} (1 - \sigma(\boldsymbol{w}_h^{\top} \boldsymbol{p}_{j,h})), & i < l+1, \\ \sigma(\boldsymbol{w}_h^{\top} \boldsymbol{p}_{query,h}), & i = l+1. \end{cases} \end{aligned} \tag{201}$$

We still use hinge loss. Therefore, the $2^H$-classification problem can be decomposed into $H$ independent binary classification problems. Our analytical technique and results for the binary classification case can then be applied. We retain only the discussion of the binary classification case in the main text and omit the detailed derivations for the multi-class setting in order to highlight the main contributions of our theoretical analysis.

## I. Extension to Linear Regression Problems

Our theoretical analysis can be extended to a linear regression problems. Note that (Huang et al., 2023) analyze the linear regression problem in the ICL framework for one-layer single-head Transformers under similar data assumptions to ours, i.e., that the data are defined by orthogonal relevant features. For Mamba, we can conduct a similar analysis. The main challenges lie in the gradient and convergence analysis under the squared loss, as well as in the formulation and analysis of outliers. One option is to formulate the context label with outliers as random outputs. With squared loss, the gradient of $\boldsymbol{W}_C$, $\boldsymbol{W}_B$, and $\boldsymbol{w}$ are computed as

$$\frac{\partial \ell(\Psi; \boldsymbol{P}^n, z^n)}{\partial \boldsymbol{W}_C} = (F(\Psi, \boldsymbol{P}) - z^n) \sum_{i=1}^{l} G_{i,l+1}^n(\boldsymbol{w}) y_i^n \boldsymbol{W}_B \boldsymbol{p}_i^n \boldsymbol{p}_{query}^n{}^\top, \tag{202}$$

$$\frac{\partial \ell(\Psi; \boldsymbol{P}^n, z^n)}{\partial \boldsymbol{W}_B} = (F(\Psi, \boldsymbol{P}) - z^n) \sum_{i=1}^{l+1} G_{i,l+1}^n(\boldsymbol{w}) y_i \boldsymbol{W}_C \boldsymbol{p}_{query} \boldsymbol{p}_i^\top, \tag{203}$$

$$\begin{aligned}
&\frac{\partial \ell(\Psi; \boldsymbol{P}^n, z^n)}{\partial \boldsymbol{w}} \\
&= (F(\Psi, \boldsymbol{P}) - z^n) \sum_{i=1}^{l} y_i^n \boldsymbol{p}_i^n{}^\top \boldsymbol{W}_B^\top \boldsymbol{W}_C \boldsymbol{p}_{query}^n G_{i,l+1}^n(\boldsymbol{w}) (\sum_{s=i+1}^{l+1} \sigma(\boldsymbol{w}^\top \boldsymbol{p}_s^n) \boldsymbol{p}_s^n \\
&\quad - (1 - \sigma(\boldsymbol{w}^\top \boldsymbol{p}_i^n)) \boldsymbol{p}_i^n).
\end{aligned} \tag{204}$$

Since $F(\Psi, \boldsymbol{P})$ is generally between $-1$ and $1$ before convergence, we can still ensure that the model can learn relevant patterns by gradient updates, which is consistent with the case of classification problem. Random labels for outlier examples cancel out their gradient contribution, leading the nonlinear gating to learn outlier patterns. Then, the further analysis is almost the same as the classification problem.

## The Use of Large Language Models

We used large-language models (ChatGPT) to help polish the writing of this paper.

